# Impact of Urban Stormwater Runoff on Cyanobacteria Dynamics in A Tropical Urban Lake

**Talita F. G. Silva** [1,*], **Brigitte Vinçon-Leite** [2], **Bruno J. Lemaire** [2], **Guido Petrucci** [3], **Alessandra Giani** [4], **Cléber C. Figueredo** [4] and **Nilo de O. Nascimento** [1]

1. Department of Hydraulic Engineering and Water Resources, Universidade Federal de Minas Gerais, 6627 av. Antônio Carlos, Belo Horizonte MG 31270-901, Brazil; niloon@ehr.ufmg.br
2. LEESU, Ecole des Ponts Paris Tech, UPEC, AgroParisTech, F-77455 Marne-la-Vallée, France; b.vincon-leite@enpc.fr (B.V.-L.); bruno.lemaire@enpc.fr (B.J.L.)
3. Prolog Ingénierie, 5-7 rue de Metz, 75010 Paris, France; guido.petrucci@gmail.com
4. Department of Botany, Institute of Biological Sciences, Universidade Federal de Minas Gerais, 6627 av. Antônio Carlos, Belo Horizonte MG 31270-901, Brazil; agiani@icb.ufmg.br (A.G.); cleberfigueredo@ufmg.br (C.C.F.)
* Correspondence: talita.silva@ehr.ufmg.br; Tel.: +55-31-34091072

**Abstract:** Worldwide, eutrophication and cyanobacteria blooms in lakes and reservoirs are a great concern for water resources management. Coupling a catchment hydrological model and a lake model has been a strategy to assess the impact of land use, agricultural practices and climate change on water quality. However, research has mainly focused on large lakes, while urban reservoirs and their catchments, especially in tropical regions, are still poorly studied despite the wide range of ecosystem services they provide. An integrated modelling approach coupling the hydrological model Storm Water Management Model SWMM and the lake ecological model DYRESM-CAEDYM is proposed for Lake Pampulha (Brazil). Scenarios of increased imperviousness of the catchment and of reduction in the load of nutrients and total suspended solids (TSS) in dry weather inflow were simulated. Runoff water quality simulations presented a fair performance for TSS and ammonium ($NH_4^+$) while the dynamics of total phosphorus (TP) and nitrate ($NO_3^-$) were poorly captured. Phytoplankton dynamics in the lake were simulated with good accuracy (Normalized Mean Absolute Error, NMAE = 0.24 and r = 0.89 in calibration period; NMAE = 0.55 and r = 0.54 in validation period). The general trends of growth, decline and the magnitude of phytoplankton biomass were well represented most of the time. Scenario simulations suggest that TP reduction will decrease cyanobacteria biomass and delay its peaks as a consequence of orthophosphate ($PO_4^{3-}$) concentration reduction in the lake surface layers. However, even decreasing TP load into Lake Pampulha by half would not be sufficient to achieve the water quality objective of a maximum concentration of 60 μg *chla* $L^{-1}$. Increased imperviousness in the catchment will raise runoff volume, TSS, TP and $NO_3^-$ loads into Lake Pampulha and promote greater cyanobacteria biomass, mainly in the beginning of the wet season, because of additional nutrient input from catchment runoff. Recovering Lake Pampulha water quality will require the improvement of the sanitation system. The lake water quality improvement will also require more sustainable and nature-based solutions for urban drainage in order to reduce non-point pollution through infiltration and retention of stormwater and to enhance natural processes, such as chemical sorption, biodegradation and phytoremediation. The integrated modelling approach here proposed can be applied for other urban reservoirs taking advantage of existing knowledge on Lake Pampulha.

**Keywords:** lake ecological modelling; SWMM; DYRESM-CAEDYM; runoff water quality; Lake Pampulha; tropical reservoirs

## 1. Introduction

Water bodies provide many valuable ecosystem services. Eutrophication is a major concern for scientists, policy-makers and citizens because of its major impact on the ecosystem functioning, in particular, the loss of biodiversity and the disruption of water uses such as drinking water supply, recreation and fishing [1]. Lentic ecosystems are especially vulnerable to eutrophication because of their high-water retention time [2] and eutrophic lakes and reservoirs are frequently affected by cyanobacteria blooms, including potential toxic species which can be harmful to human and animal health [3–5].

Many studies have investigated the impact of climate change on cyanobacteria dynamics and it is expected that the frequency and intensity of cyanobacteria blooms will increase in response to global warming [6–8]. In this paper, we investigated how increasing catchment urbanization impacts cyanobacteria dynamics in urban reservoirs. Urbanization increases pollutant emissions and leads to changes in land use and an increase of impervious areas. Increased imperviousness raises runoff in volume and speed, resulting in a greater wash-off capacity and exposing urban reservoirs to higher loads of nutrients, suspended solids, trace metals and other pollutants [9,10]. Besides, in developing countries, poor sanitation infrastructure makes municipal and industrial wastewater an important source of organic matter and nutrients to urban water bodies [11,12]. Additionally, due to small depths, urban reservoirs are frequently subject to the mixing and refilling of the water column with solids and nutrients from lower layers, which opens the way for successive cyanobacteria blooms [13].

Both urban runoff generation and the dynamics of biological processes in lentic ecosystems, particularly cyanobacteria dynamics, are complex processes which involve a large number of variables and occur on different time and space scales. The various pathways by which cyanobacteria can access nutrients from urban catchments and the processes that can stimulate or inhibit bloom formation are not yet well understood [14]. Coupling hydrological and lake ecological models is a useful approach to investigate links between catchment changes and cyanobacteria dynamics, especially in urban lakes [2,15,16]. Runoff simulated by the hydrological model is used as inflow input into a hydrodynamic-ecological lake model which simulates cyanobacteria dynamics. Such an integrated modelling approach should be well accepted by water resource managers and stakeholders because it is perceived as a holistic representation of natural systems and their interconnections, allowing them to propose and evaluate management strategies at the catchment scale [17–19].

Two or more mathematical models can be coupled to figure out how changes in the catchment can impact the water quality of lentic ecosystems and to design eutrophication control actions. But it is a relatively recent research issue and many authors agree that more integrated modelling studies must be conducted [20,21].

Different performances of urban drainage best management practices were tested in order to reduce phosphorus and nitrogen loads into a reservoir in the USA [22]; the impacts of climate change on the hydrological cycle and on phytoplankton dynamics in reservoirs were studied by Taner et al. in the USA [23], Markensten et al. [24] in Sweden and Komatsu et al. [25] in Japan. Me et al. compared the effects of best management practices in agriculture and of climate change on the nutrient load and algal biomass in a eutrophic shallow lake in New Zealand [26]. Nobre et al. explored how improved sanitation in the catchment reduced the nutrient and organic matter load and benefited Xiangshan Gang bay in China [27]. The responses of phytoplankton to changes in land and water use in the catchment were addressed by Norton et al. [18] in the UK and by Bucak et al. [28] in Turkey.

Until now, research has mainly focused on large lakes and dealt mainly with rural catchments, while reservoirs located in urban areas are still understudied [2,29,30]. These studies are mainly devoted to general water quality parameters, nutrients (N, P), oxygen or total phytoplankton biomass. Few results specifically address cyanobacteria dynamics. This paper presents a modelling strategy for assessing the link between increasing imperviousness and improvements in the sanitation system of an urban catchment and the cyanobacteria dynamics in Lake Pampulha, a tropical urban reservoir in Brazil. To our knowledge, this is the first time that the impact of increased imperviousness in an

urban catchment on cyanobacteria dynamics is assessed using a coupled model approach. In fact, in urban regions, there is a very strong need to investigate the links between lake ecological functioning and catchment changes. Urban water bodies need to be studied not as an isolated unit of the urban environment, but as a part that integrates and responds to changes occurring in its surroundings.

## 2. Materials and Methods

### 2.1. Study Site

Lake Pampulha (Figure 1 and Table 1) is a medium-sized and hypereutrophic reservoir located in Belo Horizonte city, Brazil (19°51′05.7′′ S 43°58′46.4′′ W). The main morphometric characteristics of Lake Pampulha and water quality parameters are listed in Table 1. The reservoir was built in the 1930s to supply drinking water to the city, however since the 1970s the water quality has degraded due to the rapid catchment urbanization with neither sanitation infrastructure nor erosion control. In 2003, a fluvial water treatment plant (FWTP) was installed downstream the confluence of the main tributaries of Lake Pampulha, Ressaca and Sarandi creeks (confluence of tributaries 5 and 6 in Figure 1), in order to treat dry weather flow. However, the FWTP showed little efficiency in reducing ammonium and phosphorus loads into the lake [31,32]. Nowadays, lake silting and frequent episodes of cyanobacterial blooms are the main problems to be tackled in the reservoir [32,33].

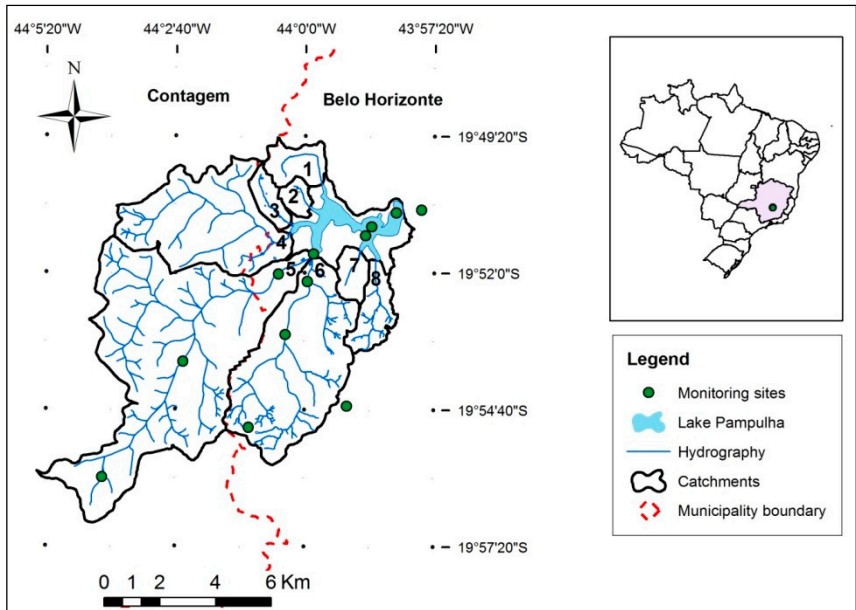

**Figure 1.** Lake Pampulha and its catchment location and tributaries: (1) Olhos d'água; (2) AABB; (3) Braúnas; (4) Água Funda; (5) Sarandi; (6) Ressaca; (7) Tijuco; (8) Mergulhão.

**Table 1.** Lake Pampulha morphological characteristics [33] and water quality from December 2011 to May 2013 (minimum value - maximum value (average value).

| Morphological Characteristics | | Water Quality | |
| --- | --- | --- | --- |
| Altitude | 801 m | TP | 58–925 (207) µg P L$^{-1}$ |
| Mean depth | 5.1 m | PO$_4^{3-}$ | 1.7–113.1 (22.1) µg P L$^{-1}$ |
| Maximum depth | 16.2 m | NH$_4^+$ | 1.4–14.8 (5.7) mg N L$^{-1}$ |
| Surface | 197 ha | NO$_3^-$ | 3.1–460 (82.9) µg N L$^{-1}$ |
| Volume | $9.9 \times 10^6$ m$^3$ | Chlorophyll-a | 19.5–322.0 (113.0) µg L$^{-1}$ |

Despite the degraded water quality, Lake Pampulha is an important tourist spot. The area surrounding the lake is used for recreational and sportive activities and it contributes to reduce flood

risk in the neighbourhood. Furthermore, from 2013 to 2016, a severe drought impacted many water supply systems in southeast Brazil, threatening water security in many large Brazilian cities, including Belo Horizonte [34]. In the perspective of an increase of the water demand by a future larger population associated to a higher frequency of extreme drought events due to climate change, Pampulha reservoir will be expected to provide water for non-drinkable uses.

The climate is characterized by a dry cool season between April and September and a wet warm season from October to March, when 90% of the total annual rainfall occurs (mean total annual rainfall is 1500 mm ) [35]. The monthly mean air temperature spreads from 18 °C in July to 23 °C in February. Lake Pampulha is fed by eight small creeks (Figure 1). Sarandi and Ressaca creeks represent 70% of total inflow and are the most polluted [36]. Lake Pampulha catchment, with a surface of 98 km$^2$ and 424,000 inhabitants, has 70% of its area urbanized, 13% occupied by vegetation, 12% by bare soil and farming and 5% by rocks and water [37].

### 2.2. Monitoring and Data Collection

#### 2.2.1. Catchment Monitoring

The monitoring of Ressaca and Sarandi sub-catchments was conducted in partnership with the municipality of Belo Horizonte (PBH) which operates a hydrometric network in the city. Rainfall heights and water levels were automatically measured every 10 min from October 2011 to June 2013. Rainfall data were provided by seven rainfall stations located in or around the catchment (Figure 1). Water level data in Sarandi and Ressaca creeks were measured by two discharge stations (Figure 1). Flow velocity was measured using a flow velocity probe (Flow Probe 1.0, Global Water, Fairborn, OH, USA) and floats, respectively, in dry weather and during rain events. Discharges were computed from water level and flow velocity data. In order to take into account the contribution of other tributaries to Lake Pampulha, Ressaca and Sarandi flow rates were increased by 30%.

Monthly or bimonthly physical-chemical monitoring of water quality in Ressaca and Sarandi creeks was conducted from February 2012 to January 2013. Samples were collected just downstream the confluence of the two creeks (Figure 1). From February to May 2013, six rain events (28 February, 15 March, 27 March, 8 April, 22 and 23 May 2013) were monitored using an automatic sampler (ISCO 3700, Teledyne ISCO, Lincoln, USA) with a water level sensor (Liquid Level Actuator, Teledyne ISCO, Lincoln, USA). Sampling started when the water level in the creek reached the water level sensor. Thereafter, 800 mL-water samples were collected every 10 min for 4 h. The samples were transported to the laboratory for filtration, storage and subsequent analyses. The following variables were determined according to APHA [38]: total suspended solids (TSS), total Phosphorus (TP), nitrate ($NO_3^-$) and ammonium ($NH_4^+$). Water temperature was measured every 5 min from February to April 2013 using a thermistor (Global Water, Fairborn, OH, USA).

A satellite image of 12 February 2013 from Google Earth software (free version 7.1.1.1888) was used to obtain land-use and imperviousness area in the catchment. Impervious area in the catchment was estimated to represent 42% of the catchment total area. The ArcGIS software (Esri, Redlands, USA, version 10.1) was used to vectorize land-use classes and to quantify the surface area associated to each of them.

#### 2.2.2. Lake Monitoring

Lake monitoring was conducted from October 2011 to June 2013. Monthly samples were collected at different depths (always at 0.5 m depth and from December 2012 also at 1.5 m, 3.5 m and 7.0 m depth) in a central site in the lake (Figure 1) and analysed for TP, orthophosphate ($PO_4^{3-}$), $NO_3^-$, $NH_4^+$ according to APHA [38]. Chlorophyll-a (hereafter *chl-a*) analysis was performed according to Nusch [39] at all sampling depths. Subsurface samples were collected with a Van Dorn sampler for phytoplankton counting and biovolume estimation through microscope analysis according to Utermöhl [40]. Vertical profiles of water temperature, conductivity, pH and dissolved oxygen were

measured every 0.5 m using an YSI multiparameter probe (YSI 556, USA). Water transparency was measured using a Secchi disk. Hourly meteorological data were provided by the nearest weather station (A521-Belo Horizonte, Pampulha) of the Brazilian National Institute of Meteorology (INMET), located 3 km from the lake. Lake bathymetry was obtained from Resck et al. [33].

## 2.3. Catchment Model

Rainfall-runoff simulation in Ressaca and Sarandi catchments was performed using the Storm Water Management Model-SWMM [41]. This is a deterministic, semi-distributed and conceptual hydrological model which also simulates associated wet weather non-point pollutant processes and flood routing in rivers and channels. In SWMM, the sub-catchments are the elements where rain falls and runoff is produced. Nodes are entry points of runoff into the drainage network which in turn is represented by the conduits. Sub-catchments are divided in pervious and impervious surfaces and infiltration occurs in the former ones. Each sub-catchment is represented by a non-linear reservoir whose outflow is computed with the kinematic wave approach. Flow routing in the channels is computed by numerical solution of Saint-Venant 1D equations and both the dynamic and kinematic wave approaches are available. More details about the model can be found in [41].

Simulation of the runoff water quality in SWMM is processed in two steps: firstly, the pollutant build-up over the sub-catchment surface during dry weather is computed with an exponential empirical Equation (1); then the pollutant wash-off during wet weather is computed with an empirical differential Equation (2). The terms of the equations are listed in Table 2.

$$M_{build-up}(t) = F_{build-up} \ x \times \left(1 - e^{-dec.t}\right) \tag{1}$$

$$M_L = w \times Q^{wpo} \times TM_{build-up} \tag{2}$$

**Table 2.** Terms of the runoff water quality model equations [41,42].

| Term | S. I. Unity | Description | Term | S. I. Unity | Description |
|------|-------------|-------------|------|-------------|-------------|
| $Q$ | mm h$^{-1}$ | Runoff rate per unit area | TM$_{build-up}$ | kg | Pollutant total mass |
| $M_{build-Up}$ | kg ha$^{-1}$ | Pollutant mass per unit area | $M_L$ | kg h$^{-1}$ | Wash-off pollutant mass per hour |
| $t$ | day | Time | $wpo$ | - | Exponent |
| $F_{build-Up}$ | kg ha$^{-1}$ | Maximum build-up per unit area | $w$ | mm$^{-1}$ | Wash-off empirical coefficient |
| $dec$ | day$^{-1}$ | Build-up rate constant | | | |

Runoff quantity modelling was performed with 57 sub-catchments and at a time step of 10 min in a previous work, providing good results [15]. The model was calibrated and validated; detailed information about model setup, calibrated parameters and calibration procedure is found in [15,43].

For the runoff water quality modelling, TSS, NO$_3^-$, NH$_4^+$ and TP concentration were selected because these are relevant pollutants for Lake Pampulha water quality. $F_{build-up}$, $dec$, $w$ and $wpo$ were calibrated using a genetic algorithm method implemented in the Matlab software (The MathWorks®, Natick, MA, USA, version 2010). The range of parameter values used in calibration is shown in Appendix A (Table A1). Calibration was performed using a time step of 10 min and with data from rain events of 28 February, 15 March and 8 April 2013 whose intensities covered the observed range during the monitoring period. Validation was performed using a time step of 10 min and the remaining observed rain events (27 March, 22 and 23 May 2013). Since runoff water quality monitoring was performed downstream of the Ressaca and Sarandi confluence, their catchments were jointly modelled. A dry weather flow was input, based on monitoring data during dry weather. TSS, NO$_3^-$, NH$_4^+$ and TP concentration in dry weather flow was respectively, 25.8 mg L$^{-1}$, 1.54 mgN L$^{-1}$, 8.5 mgN L$^{-1}$ and 1.26 mg L$^{-1}$.

Regarding the calibration of water quality parameters, tests were performed in order to define the best strategies concerning the objective function and the number of simulations for parameter calibration. Root mean squared error (RMSE, Table 3) was the most sensitive to the optimization procedure and was selected as objective function. In the genetic algorithm, 2250 simulations were performed using a population size of 50 initial parameter sets. For the whole catchment, a mean value for $F_{build-up}$, *dec* and *w* were calibrated for TSS and nutrients, whereas *wpo* was calibrated for each sub-catchment. This strategy was adopted in order to reduce parameter correlation occurrence [44]. Model performance during calibration and validation was assessed using the normalized RMSE (NRMSE, Table 3) and Pearson's correlation coefficient (r, Table 3) to compare measured and simulated concentration of TSS and nutrients. Table 3 shows all metrics which were used in this work in order to assess model performance.

**Table 3.** Metrics for assessing model performance, where n is the number of observations of y, $y_i$ is the ith observation, $\overline{y}$ is the mean of observations, $\hat{y}$ is the modelled value and $\widetilde{y}$ is the mean of modelled values [45].

| Name | Formula | Range | Ideal Value | Notes |
|---|---|---|---|---|
| Root Mean Square Error (RMSE) | $\sqrt{\frac{1}{n}\sum_{i=1}^{n}\left(y_i-\hat{y}_i\right)^2}$ | $(0, +\infty)$ | 0 | *RMSE* express the error metric in the same units as the original data. Squaring the data causes bias towards large events. |
| Normalized Root Mean Square Error (NRMSE) | $\frac{RMSE}{max(y)-min(y)}$ | $(0, +\infty)$ | 0 | *RMSE* is normalized by data range allowing comparison between study sites. |
| Normalized Mean Absolute Error (NMAE) | $\frac{\frac{1}{n}\sum_{i=1}^{n}|y_i-\hat{y}_i|}{max(y)-min(y)}$ | $(0, +\infty)$ | 0 | *NMAE* reduces the bias towards large events and allows comparison between study sites. |
| Pearson's correlation coefficient (r) | $\frac{\sum_{i=1}^{n}(y_i-\overline{y})(\hat{y}_i-\widetilde{y})}{\sqrt{\sum_{i=1}^{n}(y_i-\overline{y})^2}\sqrt{\sum_{i=1}^{n}(\hat{y}_i-\widetilde{y})^2}}$ | $(-1,1)$ | 1 | *r* measures the linear correlation of the measured and modelled values |

## 2.4. Phytoplankton Dynamics Model

Most lake ecosystem models are composed of two parts, a hydrodynamic module and an ecological module. The first one describes the physical processes of transport and turbulence in the water column, while the second one represents the main chemical and biological processes that affect phytoplankton and higher trophic levels. In this study, the deterministic models DYnamic Reservoir Simulation Model and Computational Aquatic Ecosystem DYnamics Model, DYRESM-CAEDYM (hereafter DYCD) were used to simulate the water temperature and cyanobacteria dynamics in Lake Pampulha. DYRESM is a one-dimensional hydrodynamic model that simulates the vertical distribution of temperature, salinity and density of water in lakes and reservoirs [46]. CAEDYM is composed by partial differential equations of mass conservation for different variables, such as dissolved oxygen, nutrients, suspended solids, etc. DYCD can be used in biological and/or chemical studies related to the nutrient cycle, algal succession, or the dynamics of heavy metals and pathogens in water [47]. Further information about DYCD can be found in [46,48].

### 2.4.1. Configuration, Calibration and Validation of the Phytoplankton Dynamics Model

This paper focused on cyanobacteria dynamics that was the dominant phytoplankton group during our study period [49]. In addition to the mandatory variables of water temperature, dissolved oxygen, nitrogen, phosphorus and carbon, the DYCD model was set up to explicitly simulate cyanobacteria. Phytoplankton was represented by two variables: cyanobacteria and the other phytoplankton species clustered into one single group. The processes represented in our model configuration include growth limited by water temperature, light and nutrient availability; losses related to the biological mortality, excretion and respiration; grazing, settling and resuspension. The main equations for our

phytoplankton simulation are presented in Appendix B (Tables A2 and A3) and the variables and parameters of the equations are described, respectively, in Tables A4 and A5 in Appendix B.

In this configuration, the input data required for DYCD are: lake bathymetry; daily inflows and their respective temperature and nutrient concentration; daily outflows; meteorological data (wind speed, air temperature, cloud cover, solar radiation, rainfall and vapour pressure) and initial conditions for water temperature, cyanobacteria biomass and concentration of nutrients and dissolved oxygen. Daily inflows into the lake and nutrient concentrations in inflow were obtained from the hydrological model SWMM (see Section 2.5). Daily inflow water temperature was calculated through a linear regression between available inflow water temperature measurements and daily mean air temperature (data provided by A521 INMET station). Regarding the weather forcing, cloudiness was determined from the difference between the measured solar radiation and the theoretical solar radiation computed by the method described in [50]. Modelling time step was daily.

The phytoplankton identification and biovolume assessment showed that cyanobacteria were the dominant group most of the time (see Section 3.2). Since phytoplankton biomass was highly dominated by cyanobacteria, the calibration was performed by comparing simulated phytoplankton total biomass and observed *chla* concentrations using all available data (n = 16). During calibration period, observed *chla* concentrations were available only at 0.5 m depth, except on 20 March 2012 when *chla* was also measured at 3.5 m depth. In order to identify the parameters to which Lake Pampulha model is more sensitive, a manual sensitivity analysis was carried out and allowed us to select 18 parameters which were then manually calibrated through the one-at-time method, using field data collected from October 2011 to October 2012. The calibrated parameters and their ranges are shown in Appendix C (Table A6). Validation was performed using all available data (n = 43) with observed *chla* concentrations at 0.5, 1.5, 3.5 and 7.0 m depth, between October 2012 and June 2013. The initial conditions for nutrient concentrations, total phytoplankton and cyanobacteria biomass were obtained from the closest monitoring campaign data (13 October 2011 for calibration and 26 September 2012 for validation) at a central point in the lake at 0.5 m depth. To avoid biasing the results towards the higher values, the Normalized Mean Absolute Error (NMAE, Table 3) was used in the model calibration. Pearson's correlation coefficient (r, Table 3) was also computed to model performance assessment.

*2.5. Integrating Catchment Model to Phytoplankton Dynamics Model*

After calibration and validation, the catchment model output was set up as input to the phytoplankton dynamics model according to the integrated modelling scheme represented in Figure 2: the catchment model simulates runoff quantity and water quality which is input into the lake model in order to simulate phytoplankton, including cyanobacteria dynamics. Daily inflows into Lake Pampulha were obtained through Equation (3).

$$V_{daily} = \sum_{i=1}^{n} Q_i \times \Delta t \tag{3}$$

where $V_{daily}$ is the daily inflow volume (m$^3$ day$^{-1}$), n is the number of time steps in a day, $Q_i$ is the simulated inflow at time step i (m$^3$ s$^{-1}$) and $\Delta t$ is the time step duration.

TSS, NH$_4^+$, NO$_3^-$ and TP inflow concentrations were obtained through Equation (4).

$$C_{daily} = \frac{1}{V_{daily}} \sum_{i=1}^{n} Q_i \times C_i \times \Delta t \tag{4}$$

where $C_{daily}$ is the daily mean pollutant concentration (mg L$^{-1}$ day$^{-1}$), n is the number of time steps in a day, $C_i$ is the simulated pollutant concentration at timestep i (mg L$^{-1}$) and $\Delta t$ is the time step duration.

PO$_4$$^{3-}$ concentration, the phosphorus form actually uptaken by phytoplankton, was derived from TP concentration by applying a factor 0.15, according to the literature [51] and previous work in the study site [31]. The lake model was then setup, calibrated and validated as described in Section 2.4.1.

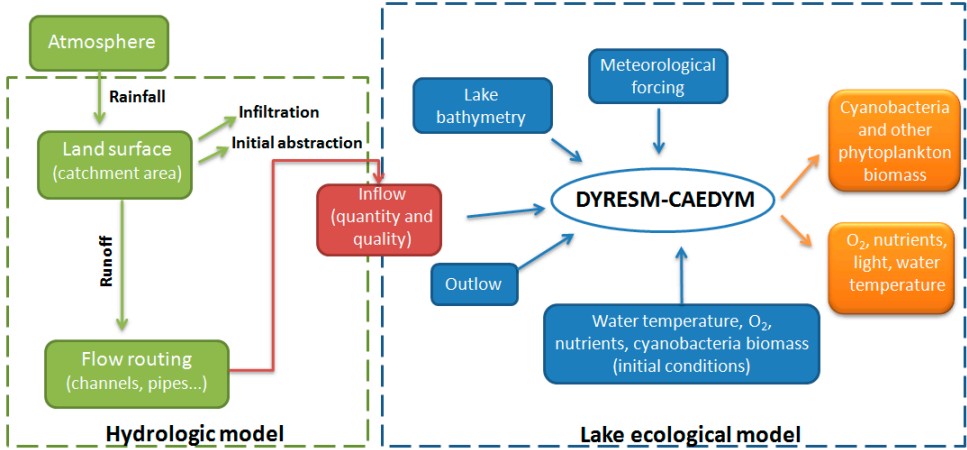

**Figure 2.** Integrated modelling approach diagram.

## 2.6. Scenarios of Catchment Changes

In 2016, Lake Pampulha and its neighbourhood have become World Heritage of the United Nations Educational, Scientific and Cultural Organization (UNESCO) and since then, greater attention has been paid to its water quality. In order to reduce point pollution into Lake Pampulha, improvement actions in the sanitation system were planned and the municipality set an objective of a maximum chlorophyll concentration of 60 μg L$^{-1}$ in the lake. However, Lake Pampulha catchment is highly urbanized and in the next years, population growth in the catchment is expected to increase impervious areas at the expense of vegetated areas.

Reducing phosphorus and nitrogen input has been pointed out by different authors as an efficient strategy to control eutrophication and cyanobacteria blooms in water bodies [52]. Improvements in the sanitation system will contribute to reduce nutrient intake in Lake Pampulha, but increasing imperviousness may raise their loads and also the load of suspended solids. Suspended solids and the associated turbidity were reported, on the one hand, to favour buoyant cyanobacteria [53] and on the other hand to impair their growth due to reduced light availability [54].

In order to obtain a first assessment of the impact of such catchment changes on cyanobacteria dynamics in Lake Pampulha, a sensitivity analysis was carried out through scenarios (Table 4): increasing sub-catchments impervious surfaces by 50% (scenario +imperviousness) and (ii) reducing by 50% the concentrations in dry weather flow of Lake Pampulha tributaries of NH$_4$$^+$ (scenario-ammonium), NO$_3$$^-$ (scenario-nitrate), TP (scenario-phosphorus) and TSS (scenario-suspended_sol), respectively. In scenario-waste_water all nutrient and TSS concentrations were reduced by 50%. With the increasing of sub-catchments impervious surfaces by 50%, catchment imperviousness increased from 42% to 63%.

Simulations were carried out for the period from October 2011 to June 2013 and the results are presented from 1 November 2011 (beginning of the wet season) to 31 October 2012 (end of the dry season) covering a hydrological year. Scenarios were compared using the following indicators based on daily values of the simulated hydrological and biogeochemical variables: (1) inflow volume, the TSS and nutrient loads into the lake and, (2) the cyanobacteria biomass averaged from 0 to 2.5 m depth and the maximum cyanobacteria biomass along the water column. The student's t-test was used to compare the mean values of variables between scenarios.

**Table 4.** Description of catchment change scenarios.

| Scenarios | Descriptions |
|---|---|
| reference | Current catchment condition |
| +imperviousness | +50% of imperviousness |
| -waste_water | $-50\%$ in TSS, $NH_4^+$, $NO_3^-$ and TP concentration in dry weather flow |
| -ammonium | $-50\%$ in $NH_4^+$ concentration in dry weather flow |
| -nitrate | $-50\%$ in $NO_3^-$ concentration in dry weather flow |
| -phosphorus | $-50\%$ in TP concentration in dry weather flow |
| -suspended_sol | $-50\%$ in TSS concentration in dry weather flow |

## 3. Results

This section firstly presents the results obtained through the catchment water quality modelling for TSS, $NH_4^+$, $NO_3^-$ and TP. Then, the results of phytoplankton dynamics provided by our integrated modelling approach are presented. Finally, the impact of catchment change scenarios on the inflow volume, the TSS and nutrient loads into the lake and on cyanobacteria dynamics are shown.

### 3.1. Runoff Water Quality Model

The results of runoff water quality modelling in the combined Ressaca and Sarandi sub-catchments are presented in Figure 3 and Table 5. Parameter calibrated values are shown in Appendix A (Table A1). As a general observation, the model was unable to represent the high variability of observed TSS and nutrient concentrations (Figure 3). The best results were obtained for TSS whose NRMSE is 0.19 in calibration and 0.24 in validation. However, the model underestimated TSS concentrations most of the time (Figure 3).

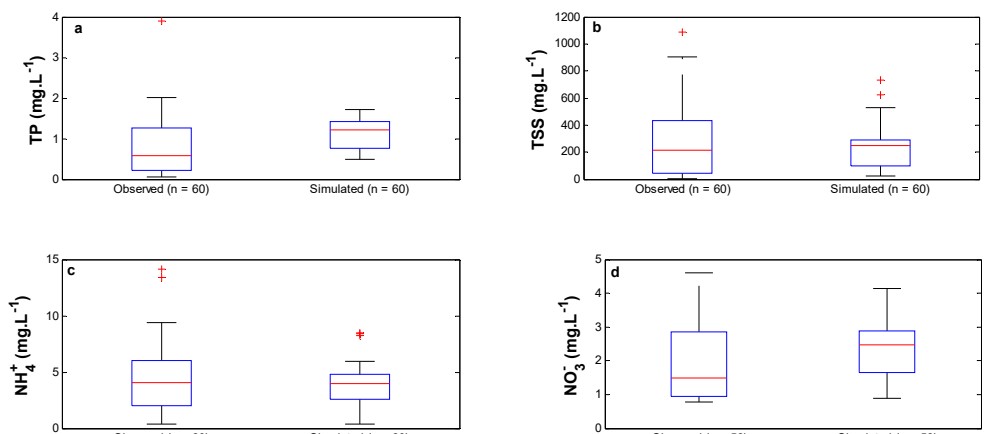

**Figure 3.** Runoff water quality in the Ressaca-Sarandi catchment: measured and simulated concentrations for (**a**) TP, (**b**) TSS, (**c**) $NH_4^+$ and (**d**) $NO_3^-$. The horizontal red line is the median value; the blue box represents the interquartile range (IQR), the whiskers the lowest value still within 1.5 IQR of the lower quartile and the highest value still within 1.5 IQR of the upper quartile and red crosses the outliers.

According to measured data, $NH_4^+$ was mainly diluted during rain events. Probably due to misconnections in the sewage system, dry weather flow was its main source. Preliminary calibration tests confirmed that runoff was a minor source of NH4$^+$ in Sarandi and Ressaca sub-catchments and thus, NH4$^+$ simulation was set to take into account only dilution of dry weather flow by stormwater, i.e., the build-up and wash-off processes were not simulated. NRMSE for $NH_4^+$ is 0.35 in calibration and 0.28 in validation.

The model was not able to reproduce TP and $NO_3^-$ concentrations at a 10-min time step. For quite a large range of observed concentrations, simulated concentrations showed minor variations

(Figure 3). TP concentrations were overestimated in most cases with NRMSE = 0.19 in calibration and 0.60 in validation. $NO_3^-$ concentrations were underestimated with NRMSE = 0.38 in calibration and 0.49 in validation.

**Table 5.** Model performance for TSS, $NH_4^+$, $NO_3^-$ and TP concentration modelling. n: number of data; NSE: Nash-Sutcliffe model efficiency, r: Pearson's correlation coefficient, NRMSE: normalized root mean square error. Cal and Val: calibration and validation periods.

| Performance | TSS | | $NH_4^+$ | | $NO_3^-$ | | TP | |
|---|---|---|---|---|---|---|---|---|
| | Cal | Val | Cal | Val | Cal | Val | Cal | Val |
| n | 29 | 31 | 29 | 31 | 29 | 27 | 29 | 31 |
| r | 0.77 | 0.18 | 0.70 | 0.61 | 0.24 | −0.36 | 0.54 | −0.60 |
| RMSE (mg L$^{-1}$) | 174 | 254 | 1.87 | 3.82 | 1.15 | 1.79 | 0.70 | 0.91 |
| NRMSE | 0.19 | 0.24 | 0.35 | 0.28 | 0.38 | 0.49 | 0.19 | 0.60 |

## 3.2. Phytoplankton Dynamics Modelling

Phytoplankton counting and volume estimation of 23 samples allowed us to verify that cyanobacteria were dominant for most of the study period. Cyanobacteria biomass represented less than 85% of total phytoplankton only on March 2012, August 2012, September 2012, May 2013 and June 2013 (Figure 4).

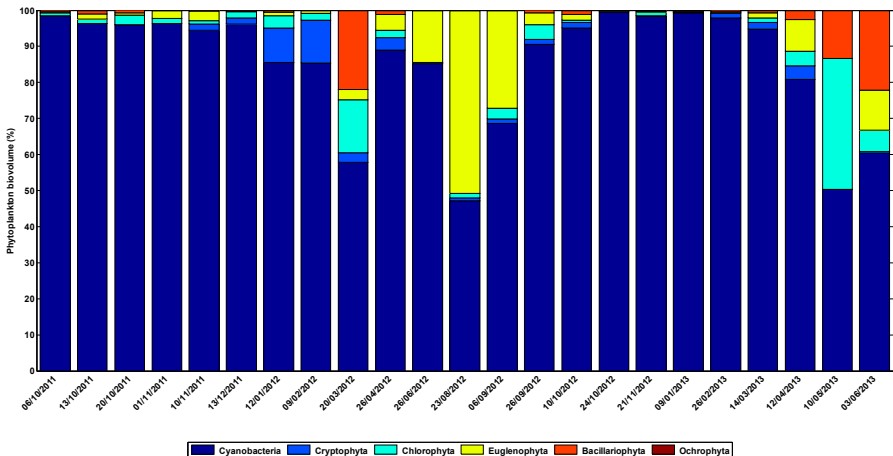

**Figure 4.** Phytoplankton composition in Lake Pampulha.

Simulated phytoplankton total biomass at 0.5 m depth is compared to observed *chla* concentrations at 0.5 m depth in Figure 5. Model performance metrics are shown in Table 6. Model performance during the calibration period (October 2011–October 2012) provided good results (NMAE = 0.24, r = 0.89, n = 16). For the validation period (October 2012–June 2013), the model performance was slightly worse (NMAE = 0.55, r = 0.54, n = 43).

The lake model reproduced the general trends of growth and decline of phytoplankton biomass with an acceptable accuracy (Figure 5a). According to model results, cyanobacteria dominated phytoplankton during the whole study period and the biomass of other phytoplankton was negligible. Total phytoplankton biomass was underestimated from September to October 2012 and overestimated in May 2013. These periods coincide with those in which cyanobacteria was less dominant (Figure 4) and showed that our model was unable to represent other phytoplankton groups dynamics. During the rest of the period, the model was able to well represent the magnitude of phytoplankton biomass.

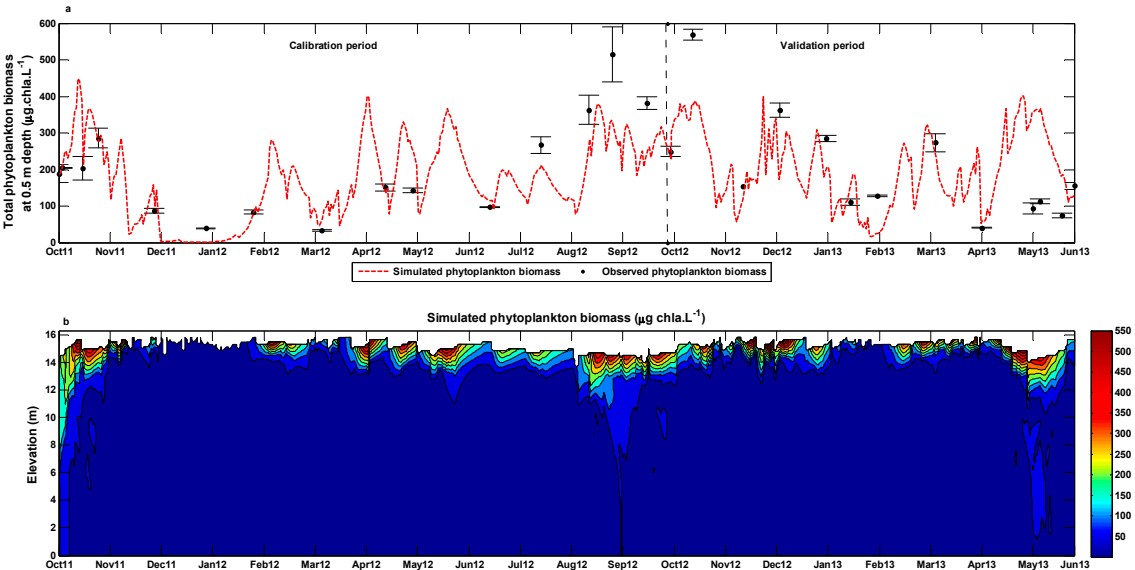

**Figure 5.** (**a**) Observed (dots) and simulated (dotted line) total phytoplankton biomass in Lake Pampulha at 0.50 m depth. Error bars indicate the standard deviation between samples replicates. The vertical dotted line separates the calibration an validation periods. (**b**) Simulated phytoplankton biomass in Lake Pampulha during the study period.

**Table 6.** Model performance for phytoplankton biomass at 0.5 m depth. NMAE is the normalized mean absolute error, r is the Pearson's correlation coefficient, and n is the number of data.

|  | NMAE | r | n |
|---|---|---|---|
| **Calibration** | 0.24 | 0.89 | 16 |
| **Validation** | 0.55 | 0.54 | 43 |

Phytoplankton biomass in Lake Pampulha can reach high values: up to 570 µg *chla* $L^{-1}$ according to monitoring data on 23 October 2012 and 562 µg *chla* $L^{-1}$ according to model results on 29 October 2011 (Figure 5b), in both cases at 0.5 m depth. Phytoplankton biomass is distributed along the first 2.5 m below the water surface (Figure 5b) because the absence of light prevents phytoplankton growth at greater depths (mean Secchi depth during the study period = 0.37 m).

### 3.3. Scenarios of Catchment Changes

Scenarios described in Table 4 were simulated and the results are presented in Figures 6–8. In order to assess hydrological indicators, only scenarios reference, +imperviousness and -waster_water were used since in the catchment model, the latest-mentioned is a linear combination of scenarios -ammonium, -nitrate, -phosphorus and -suspended_sol. Figure 6a shows the daily inflow into Lake Pampulha. Duration curves are plotted in Figure 6b for daily inflow and in Figure 7 for daily nutrient and TSS loads. Simulated monthly loads of $NH_4^+$, $NO_3^-$, TP and TSS entering into Lake Pampulha are shown in Figure 8 for scenarios reference and -waste_water.

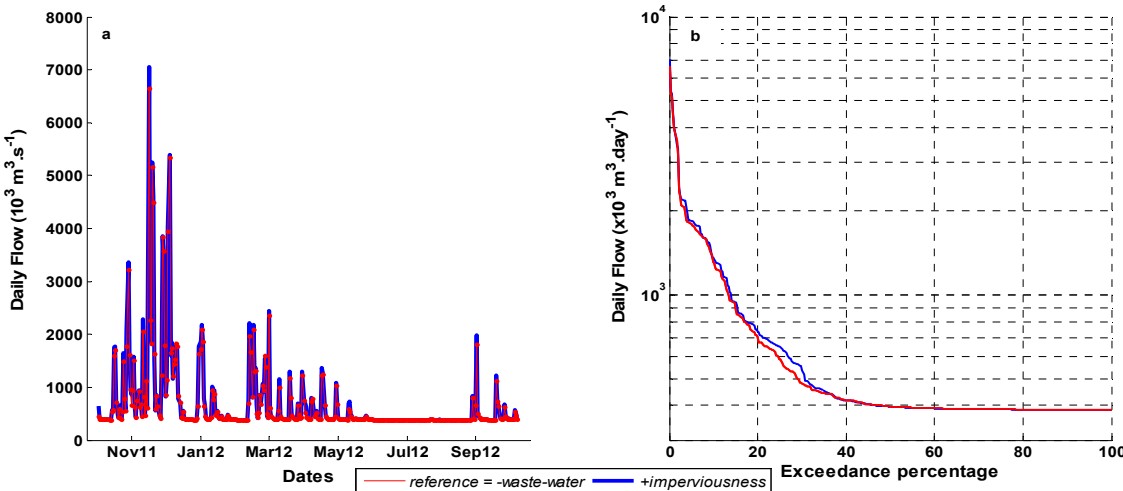

**Figure 6.** (**a**) Hydrogram for scenarios reference and -waster_water (red line), +imperviousness (blue line). (**b**) Daily flow duration curves for scenarios reference and -waster_water (red continuous line) +imperviousness (blue continuous line). In (**b**) the scale of the vertical axis is logarithmic and the plateau at high exceedance percentage corresponds to dry weather periods.

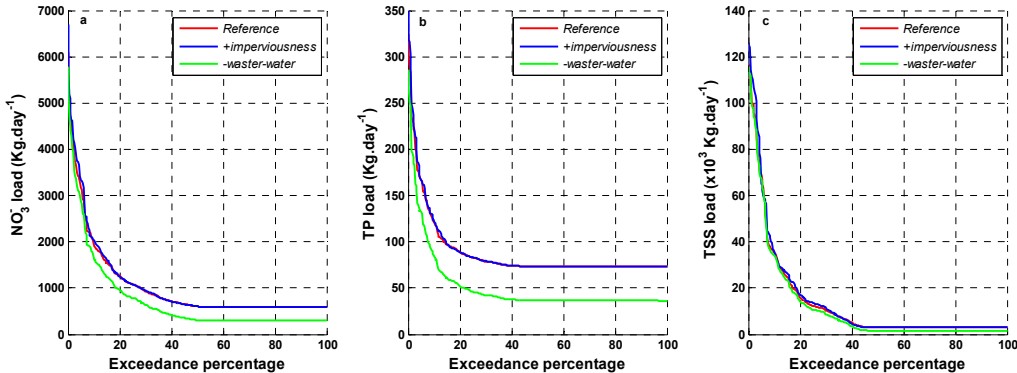

**Figure 7.** Duration curves for scenarios reference (red line), +imperviousness (blue line) and -waster_water (green line) for $NO_3^-$ (**a**), TP (**b**) and TSS (**c**). The plateau at high exceedance percentage corresponds to dry weather periods.

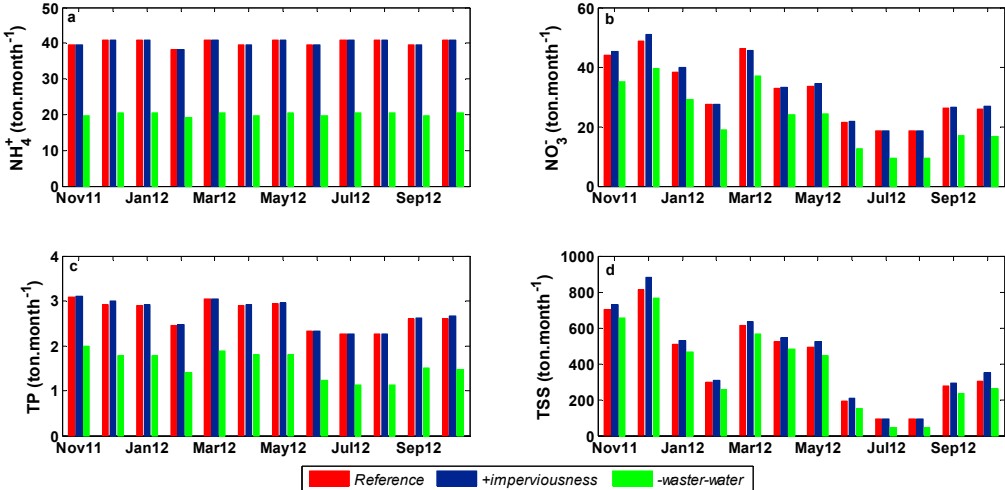

**Figure 8.** Monthly simulated loads of $NH_4^+$ (**a**), $NO_3^-$ (**b**), TP (**c**) and TSS (**d**) entering into Lake Pampulha in scenarios reference, +imperviousness and -waster_water.

The daily flow volumes of scenarios reference and -waster_water coincide, since only the quality of dry weather flow differs, not its quantity (Figure 5). In Figure 6b, the daily inflow into the lake is plotted against the percentage of time that a particular inflow was exceeded (flow duration curve). Greater inflows result from more intense rain events; they are less frequent and thus, associated to smaller exceedance percentages. Oppositely, smaller inflows are more frequent and thus associated to greater exceedance percentages. Small and more frequent flows which occur in dry weather are not impacted in scenario +imperviousness since our model configuration does not account for links between surface and groundwater flows, such as aquifer recharge. Increased imperviousness increased inflows into Lake Pampulha with exceedance percentage smaller than 35% which corresponds to flow peaks on Figure 6a.

Daily $NH_4^+$ load (data not shown) into Lake Pampulha is the same (1315 kg day$^{-1}$) in scenarios reference and +imperviousness because build-up and wash-off processes were not simulated for $NH_4^+$: its load into Lake Pampulha only relies on dry weather flow. In this way, when $NH_4^+$ concentration is reduced by 50% in dry weather inflows, an equivalent reduction occurs in the daily load into the lake.

For $NO_3^-$ (Figure 7a), TP (Figure 7b) and TSS (Figure 7c), stormwater events greatly increased the load into Lake Pampulha in all considered scenarios. The daily TSS load represents 60 times the dry weather load, $NO_3^-$ and TP loads 10 and 4 times, respectively. In scenario +imperviousness, increased runoff from the more impervious catchment led to a slight increase in $NO_3^-$ (Figures 7a and 8b) and TP (Figures 7b and 8c) daily loads for exceedance percentage below 20% and below 30% for TSS (Figure 7c) daily loads.

Most of the time, however, scenarios reference and +imperviousness coincide. The difference between $NO_3^-$, TP and TSS daily loads produced in scenarios -waste_water and reference (or +imperviousness) reduces when exceedance percentages decrease, i.e., the effect of reducing pollutant in dry weather flows is lost as more non-point pollution comes from stormwater.

Since in the phytoplankton dynamics modelling, cyanobacteria widely dominate the total phytoplankton biomass, we present the impact of catchment changes only on cyanobacteria dynamics. The daily cyanobacteria biomass averaged from 0 to 2.5 m depth is presented in Figure 9 for all scenarios. This allows us to explore the impacts of catchment changes on cyanobacteria dynamics in a broader view. The scenarios with increased catchment imperviousness (+imperviousness) and with a reduction of TP concentration (-phosphorus) and a reduction of all pollutant concentrations (-waste_water) in dry weather flow stand out from the others. Scenarios +imperviousness and -phosphorus are now compared to scenario reference and further explored for cyanobacteria biomass on Figure 10a and for $PO_4^{3-}$ concentration in Figure 10b.

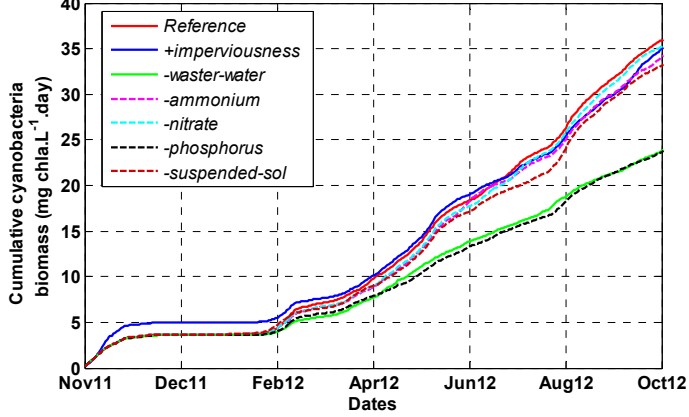

**Figure 9.** Daily cumulative cyanobacteria biomass (averaged from 0 to 2.5 m depth) in Lake Pampulha for all scenarios.

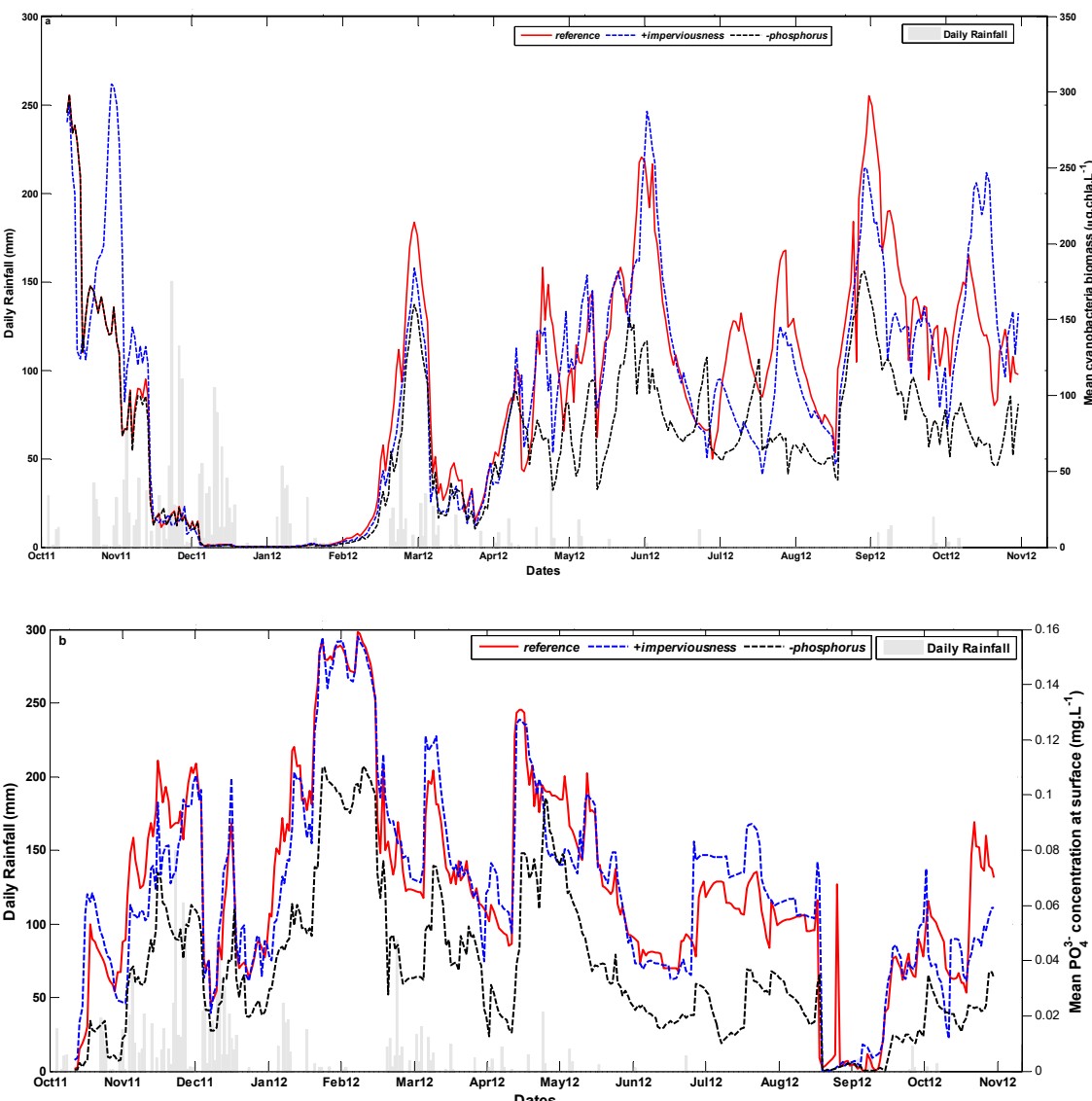

**Figure 10.** Cyanobacteria biomass (**a**) and $PO_4^{3-}$ concentration (**b**) averaged from 0 to 2.5 m depth in Lake Pampulha for scenarios reference (red continuous line), +imperviousness (dotted blue line) and -phosphorus (dotted black line). Grey bars are daily rainfall in Lake Pampulha catchment.

In the beginning of the simulation period, cyanobacteria biomass was higher in scenario +imperviousness when compared to other scenarios which correspond to the reference scenario in the same period (Figure 9). The model suggests that increasing runoff on the catchment immediately promotes an increase in cyanobacteria biomass, as on the results for November 2011 (Figure 10a). This may be related to the greater nutrient load in runoff of scenario +imperviousness. To illustrate this assumption, daily rainfall, flow, $PO_4^{3-}$ and TSS load into Lake Pampulha from 25 October to 29 November 2011 are presented in Figure 11. In-lake variables averaged from 0 to 2.5 m depth which were significantly and strongly correlated to mean cyanobacteria biomass during this period are also presented.

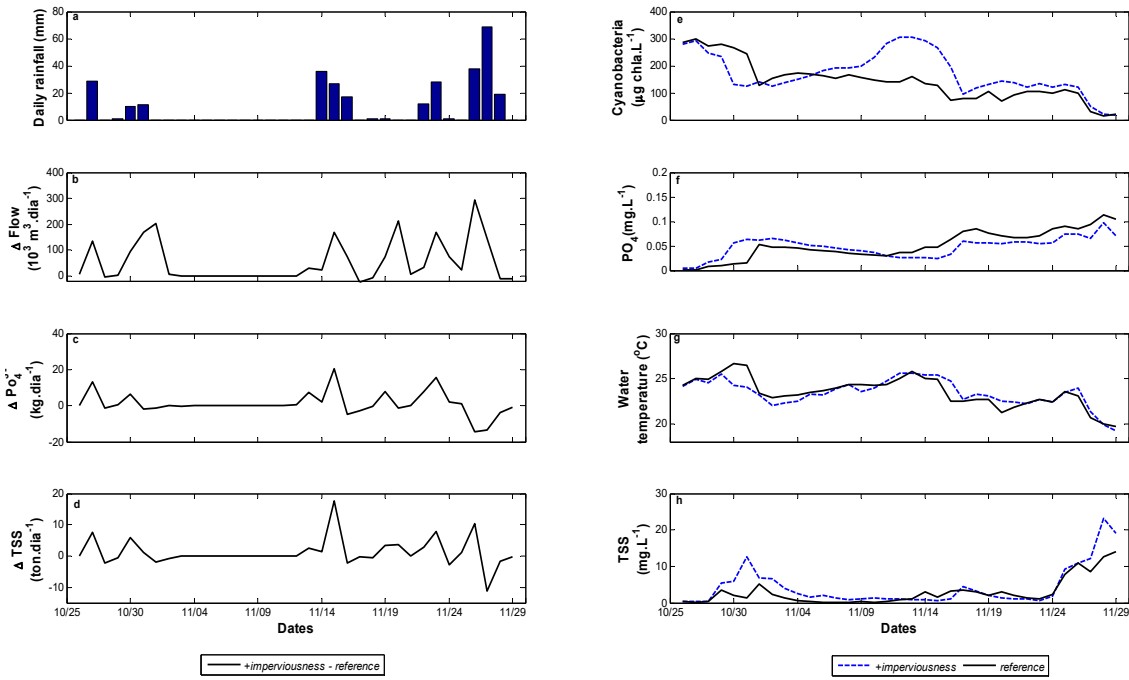

**Figure 11.** From 25 October to 29 November 2011: (**a**) Daily rainfall in Lake Pampulha catchment; difference between +imperviousness and reference scenarios for daily flow (**b**); daily $PO_4^{3-}$ load (**c**); daily TSS load (**d**); mean value (averaged from 0 to 2.5 m depth) in Lake Pampulha for scenarios reference (black continuous line) and +imperviousness (dotted blue line) of (**e**) cyanobacteria biomass; (**f**) $PO_4^{3-}$ concentration; (**g**) water temperature and; (**h**) TSS concentration.

On most rainy days, inflow (Figure 11b), $PO_4^{3-}$ (Figure 11c) and TSS (Figure 11d) loads in scenario +imperviousness are greater than in scenario reference. On 25 October 2011, cyanobacteria biomass in both reference and +imperviousness scenarios was similar. From October 27 to 31, 2011, 50 mm of rain fell, increasing lake concentrations in $PO_4^{3-}$ (Figure 11f) and TSS (Figure 11h) in scenario +imperviousness compared to scenario reference. Also, in +imperviousness, cyanobacteria biomass (Figure 11e) and water temperature (Figure 11g) were smaller due to higher TSS concentration which prevented light penetration into the water column. The rain stopped and $PO_4^{3-}$ and TSS concentration decreased in both scenarios while cyanobacteria biomass increased in +imperviousness scenario. As cyanobacteria increased, $PO_4^{3-}$ concentration decreased and from the peak of cyanobacteria biomass on 11 November 2011, $PO_4^{3-}$ became smaller in scenario +imperviousness than in scenario reference. The rain started again on 20 November 2011, but $PO_4^{3-}$ concentration in scenario +imperviousness remained smaller than in scenario reference. Increasing TSS concentration and decreasing water temperature in the end of November leaded to cyanobacteria biomass decrease in both scenarios. During this period in scenario +imperviousness, Pearson's correlation coefficients between mean cyanobacteria biomass and mean water temperature, and mean $PO_4^{3-}$ and TSS concentration were, respectively, 0.88, −0.92 and −0.70 (p-value < 0.00001).

From mid-November 2011 to the end of January 2012, no increase in cyanobacteria biomass in Lake Pampulha was observed in any of the scenarios (Figure 9). From February 2012, cyanobacteria biomass started growing in all scenarios but slower in scenarios -phosphorus and -waste_water indicating that the phosphorus concentration reduction in dry weather flow had greater impact on cyanobacteria dynamics than the reduction of $NH_4^+$ or $NO_3^-$ (Figure 9). Cyanobacteria biomass is smaller in scenario -phosphorus and its peaks seemed to occur later compared to scenarios reference and +imperviousness (Figure 10a), due to smaller $PO_4^{3-}$ concentration in surface water (Figure 10b). Most of the time, $PO_4^{3-}$ concentration in surface water is higher than 0.04 mgP L$^{-1}$ even during dry weather, in both scenarios reference and +imperviousness (Figure 10b). This shows that dry weather

flow is actually an important source of phosphorus to Lake Pampulha. In general, cyanobacteria biomass increase is followed by $PO_4^{3-}$ concentration decrease in all three scenarios (Figure 10b). In scenario -phosphorus, $PO_4^{3-}$ concentration is greatly reduced compared to scenarios reference and +imperviousness. Mean $PO_4^{3-}$ and TP concentrations in surface layer (from 0 to 2.5 m depth) dropped by half and are significantly smaller in scenarios -waster_water and -phosphorus in comparison to scenario reference (p-value < 0.00001).

Increased catchment imperviousness started to have a smaller impact on cyanobacteria dynamics from March 2012. During dry weather period in 2012 the highest cyanobacteria biomass is most often observed in scenario reference (Figure 10a). Increased catchment imperviousness had greater impact on cyanobacteria biomass also on October 2012 which corresponds to the beginning of the next wet season. However, $NO_3^-$ and $NH_4^+$ concentration seemed to play a more important role on cyanobacteria biomass than water temperature, $PO_4^{3-}$ and TSS concentration. Pearson's correlation coefficients between cyanobacteria biomass and $NH_4^+$ and $NO_3^-$ were, respectively, $-0.95$ and $0.50$ (p-value < 0.005) in scenario +imperviousness from 18 September to 18 October 2012.

The distribution of daily maximum cyanobacteria biomass was compared for all simulated scenarios (Figure 12). Scenario reference presents the highest median value of cyanobacteria biomass peak while scenario +imperviousness exhibits the largest variance and highest daily maximum values of cyanobacteria biomass. Mean cyanobacteria biomass peak is significantly smaller in scenarios -waste_water and -phosphorus than in scenario reference (p-value < 0.001) and its variance is also smaller in these scenarios. Cyanobacteria biomass peaks have quite similar distribution in scenarios reference, –ammonium, -nitrate and -suspended_sol.

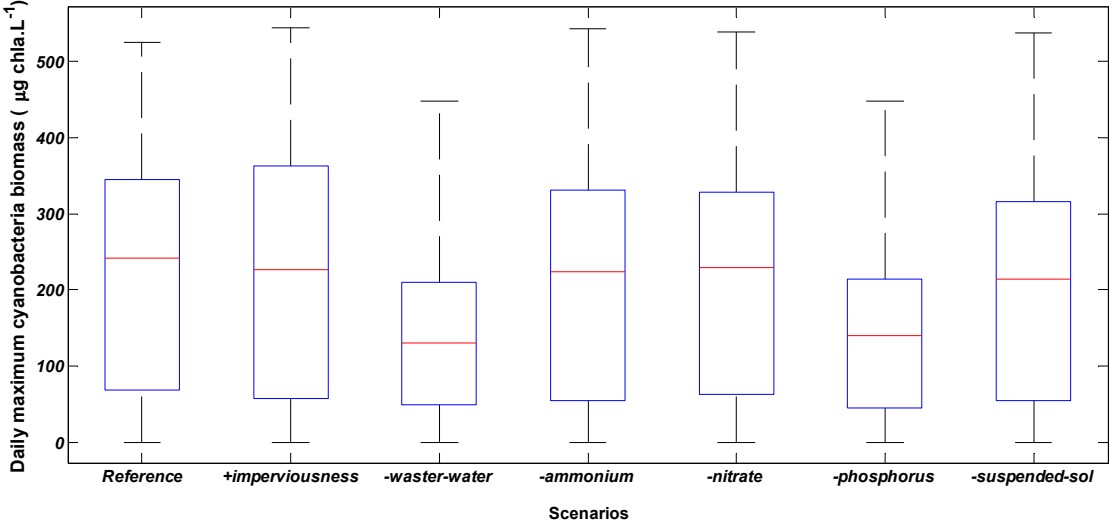

**Figure 12.** Daily maximum cyanobacteria biomass variation from 1 November 2011 to 31 October 2012 in Lake Pampulha for scenarios reference, +imperviousness, -waste_water, -ammonium, -nitrate, -phosphorus and -suspended_sol. The horizontal red line is the median value; the blue box represents the interquartile range (IQR), the whiskers the lowest value still within 1.5 IQR of the lower quartile and the highest value still within 1.5 IQR of the upper quartile and red crosses the outliers.

## 4. Discussion

In this section, we first compare the performance of our coupled catchment and lake models with literature results. We then discuss how the modelling chain simulates the impact of catchment changes on cyanobacteria dynamics. Finally, we present some perspectives for mitigation strategies.

### 4.1. Integrated Modelling Performance

Modelling runoff water quality is a very challenging task compared to runoff quantity modelling. Indeed, previous studies have demonstrated that the calibration of runoff water quality model may be challenging due to model insensitivity to optimization, obtaining several equivalent optimal parameter sets and correlations among parameters [55–57]. In the literature, relative errors around 20–30% are considered satisfactory results for stormwater pollutant load at rain event timescale [57,58]. Tsihrintzis et al. [59] applied SWMM in four catchments in the USA to simulate TSS and total Kjeldahl nitrogen (TKN) in stormwater. Pearson's correlation coefficients were 0.98 and 0.69, respectively, for TSS and TKN. Chow et al. [60] used SWMM to simulate TSS and TP in stormwater from three catchments in Malaysia. NRMSE varied from 0.06 to 0.38 for TSS and from 0.07 to 0.13 for TP. Pearson's correlation coefficient was around 0.9 for both TSS and TP. Both American and Malaysian catchments had surface area smaller than 1 km$^2$ and model performance was computed by comparing observed and simulated loads during rain events.

In this work, the hydrological model was assessed through comparison between observed and simulated pollutant concentration every 10 min. Our results for TSS and $NH_4^+$ (NRMSE varied from 0.19 to 0.35 and r from 0.18 to 0.77) were within the reasonable performance range as stated by literature. $NO_3^-$ and TP results (NRMSE varied from 0.19 to 0.60 and r from −0.60 to 0.54) were worse than reported in previous works. However, Lake Pampulha catchment has a much larger area (98 km$^2$) than catchments usually reported in studies of runoff water quality modelling and at 10-min time step, a good performance is difficult to achieve.

The spatial and temporal variability of build-up and wash-off processes at the catchment scale is likely the primary cause for the poor performance of runoff quality model in Lake Pampulha catchment and elsewhere [44,61]. Deposition and resuspension in the drainage network, erosion of pervious surfaces and erosion in streams which are not taken into account in the model may also play a key role [62,63]. Furthermore, Liu et al. [64] argue that wash-off process should not be modelled as a continuous function of rain intensity and duration. It has long been acknowledged that stormwater quality modelling has to be improved and developed. The methods used to simulate contaminant generation (build-up and wash-off process in this study) rely on empirical parameters which reflect our limited knowledge about stormwater quality generation [63,65]. The storage and release of pollutants from sediments (especially in streams) are not (or are poorly) modelled [62]. Although in many urban experimental catchments systematic monitoring of diffuse wet weather pollution sources and dynamics has lately been performed, the amount of data available is still a challenge in modelling the runoff water quality in the urban environment [57,66,67]. Finally, the calibration of model parameters may also be an important source of uncertainty, since it is affected by the objective function, the calibration method and by non-representative data which transfer their limitations to the parameters [55,56].

Despite all difficulties to accurately simulate runoff water quality, at a small time step, in Lake Pampulha catchment, especially for TP and $NO_3^-$, their mean monthly simulated loads (scenario reference), respectively 2.7 ton month$^{-1}$ and 31.8 ton month$^{-1}$, were in the same order of magnitude of previous work in this study site. Torres et al. conducted a monthly sampling program from January to December 1998 in Lake Pampulha tributaries [36]. In the confluence of Ressaca and Sarandi creeks, they estimated, respectively in wet and dry months, a TP load of 6- and 3-ton month$^{-1}$ and the load of dissolved inorganic nitrogen was estimated to 41- and 18-ton month$^{-1}$. Furthermore, the lake model has a much coarser time step (1 day) than the hydrological model (10 min), which may compensate the limitations of the latter. It should be noticed that this model configuration actually reflects the timescales of the real system: the catchment reacts rapidly to rain events, while the lake has a significant inertia. These outcomes validate the application of the integrated modelling approach proposed here, though we are aware of the limitations caused by all involved uncertainties.

Lake Pampulha model was generally able to reproduce the timing and magnitude of phytoplankton biomass during the study period. Comparing the performance of phytoplankton dynamics models is a difficult task because the performance indicators adopted and the duration of simulations vary between

studies. In a broad review of phytoplankton dynamics modelling including 124 studies, Shimoda and Arhonditsis [30] reported poor and highly variable model performance for total phytoplankton simulation: the coefficients of determination varied from 0.01 to 0.92 with a median value of 0.28; normalized relative absolute errors ranged from 0.12 to 1.41, with a median value of 0.39. More literature results are also summarized in Table 7. Comparatively, our model performance (phytoplankton biomass at 0.5 m depth: NMAE = 0.24 and r = 0.89 in calibration; NMAE = 0.54 and r = 0.54 in validation) is in agreement with the best results reported. However, the model underestimated the peak value of phytoplankton biomass from the end of August to October 2012. It may be due to a change in phytoplankton group dominance, which could not be properly captured by the model. Other phytoplankton groups besides cyanobacteria were aggregated into one group and the calibrated parameters reflect an average of assemblages of species in the calibration period, when cyanobacteria were highly dominant. Indeed, taxonomic analysis in Lake Pampulha showed that cyanobacteria biomass accounted for at least 85% of the phytoplankton community in the study period, except between late August and early September 2012 and May and June 2013 when other phytoplankton had remarkably increased [49].

**Table 7.** Model performance for phytoplankton biomass simulation. NMAE is the normalized mean absolute error, r is the correlation coefficient, RE is the mean relative error, $R^2$ is the determination coefficient. Cal: calibration period; val: validation period.

| Water Body (Country) | Maximum Depth (m) | Model | Model Performance for Total Phytoplankton Biomass (chla-a) | Reference |
|---|---|---|---|---|
| Okareka (New Zealand) | 33.5 | DYRESM-CAEDYM | RE: 0.44 (cal); 0.28 (val) $R^2$: 0.054 (cal); 0.45 (val) | [68] |
| Rotoehu (New Zealand) | 13.5 | DYRESM-CAEDYM | RE: 0.58 (cal); 0.57 (val) $R^2$: 0.004 (cal); 0.00 (val) | [68] |
| Ellesmere (New Zealand) | 2.5 | DYRESM-CAEDYM | RE: 0.31 (cal); 0.61 (val) $R^2$: 0.096 (cal); 0.18 (val) | [68] |
| Shahe reservoir (China) | 14 | DYRESM-CAEDYM | RE: 0.52 (cal); 0.73 (val) $R^2$: 0.57 (cal); 0.21 (val) | [69] |
| Lake Maggiore (Italy) | 370 | GLM-AED2 | r: 0.69 (cal); 0.41 (val) NMAE: 0.38 (cal); 0.39 (val) | [70] |
| Lake Pampulha (Brazil) | 16 | DYRESM-CAEDYM | r: 0.88 (cal); 0.74 (val) NMAE: 0.26 (cal); 0.43 (val) | This study |

### 4.2. Catchment Changes Impact on Cyanobacteria Dynamics

Despite the uncertainties related to each model individually and to their association, the sensitivity analysis performed in this study together with outcomes reported in the literature allowed us to delineate the main impacts of urban stormwater on cyanobacteria dynamics in Lake Pampulha. Both increased imperviousness in the catchment and nutrient reduction concentration in dry weather flow, mainly due to TP reduction, impacted cyanobacteria growth in Lake Pampulha.

The results of our simulated scenarios indicated that reducing TP load from dry weather flow, which is largely impacted by irregular wastewater disposal within the catchment, plays a key role in reducing cyanobacteria biomass in Lake Pampulha. Nitrogen concentration in Lake Pampulha during the study period was very high, especially $NH_4^+$, (5.7 mg N $L^{-1}$) and phosphorus limited cyanobacteria growth. Lower $PO_4^{3-}$ concentration in the water column in scenarios -waster_water and –phosphorus limited the build-up of cyanobacteria biomass leading to smaller and delayed cyanobacteria peaks.

The relationship between cyanobacteria biomass and TP in Lake Pampulha had already been highlighted in previous studies [32,36]. Figueredo et al. [32] observed a long-term continuously increasing trend for TP from 1996 to 2011 which was identified as a major driver of persistent blooms of cyanobacteria while no significant correlation was found between cyanobacteria and $NO_3^-$ nor with $NH_4^+$. According to the authors, this TP increasing trend may be related to increasing wastewater disposal in Lake Pampulha tributaries and the associated inefficiency of the fluvial waste water treatment plant. They also assumed that internal phosphorus release from the sediments could be an important phosphorus source.

Phosphorus has long been recognized as a main trigger of lake eutrophication [71] and as the main limiting factor for the growth of cyanobacteria in many lakes and reservoirs. However, even with a reduction in TP load by half, cyanobacteria biomass still reached values much larger than 60 $\mu$g *cha* L$^{-1}$, the intended water quality goal for Lake Pampulha. It is also important to note that reducing phosphorus external loads may be insufficient to control eutrophication and cyanobacteria blooms in lentic systems because phosphorus can release from the sediments under anoxic conditions (internal phosphorus load) and promote cyanobacteria blooms in surface layers after mixing events [72]. Finally, the role of nitrogen on cyanobacteria growth may not be discarded because in an environment with such high levels of nitrogen, the management of phosphorus only may be insufficient to avoid cyanobacteria dominance [52,53,73].

Our results suggest that besides dry weather flow, urban stormwater is also an important source of TP, $NO_3^-$ and TSS to Lake Pampulha. The increasing imperviousness of the catchment raises inflow volume and nutrient loads into Lake Pampulha. Greater stormwater volume from a more impervious catchment seems to promote cyanobacteria growth in Lake Pampulha, especially in the beginning of the wet season when $PO_4^{3-}$ and $NO_3^-$ concentration in the lake surface layer is low. During dry weather, $PO_4^{3-}$ and $NO_3^-$ become less available in the surface layers as a consequence of cyanobacteria uptake. With the beginning of the rain period, additional nutrient inputs from catchment runoff may provide favourable conditions for cyanobacteria blooms if the TSS input is not too high. Indeed, in Lake Pampulha, a high TSS concentration in the surface layers prevents light penetration and may delay cyanobacteria growth or even contribute to its decrease.

The impact of increasing urban imperviousness on cyanobacteria dynamics in urban reservoirs, as far as we know, is rarely reported in literature. The predicted impacts of increasing rainfall intensities due to climate change or the observed impacts of extreme rainfall events on cyanobacteria and other phytoplankton dynamics are well documented [6,74–76]. In the large and shallow Lake Taihu (China), typhoons in 2011 and 2012 increased turbidity, as well as $NH_4^+$, $PO_4^{3-}$, TP and total nitrogen (TN) concentration in the water column leading to a bloom of *Microcystis* spp. The influence of typhoons on nutrient concentrations persisted for a long time (more than two weeks) [75]. In Lake Okeechobee (USA), hurricanes promoted high turbidity in the water column leading to light limitation and a decline in phytoplankton biomass concomitant with a higher availability of nitrogen and phosphorus in the water column. When suspended solids settled, cyanobacteria took advantage of the high availability of nutrients and a particular intense bloom was observed in Lake Okeechobee [54]. On the opposite, in Lake George (USA) where cyanobacteria are favoured by low discharges and low water turnover, their biomass was significantly lower during periods of high rainfall and water levels (El Niño years), while TP and colour were significantly higher [54].

Increasing imperviousness in the catchment or increasing rain intensities due to climate change are expected to lead to higher runoff volumes into water bodies with greater nutrient loads, especially in particulate form [51,77]. Higher inflows and loads of nutrients and suspended solids into reservoirs may have site specific effects on cyanobacteria dynamics which may also vary according to the timescale, as suggested by our results and by literature [6,74]. In the short term, increased inflow volumes may cause higher flushing rates, increase turbulence and water column mixing, leading to a reduction of cyanobacteria biomass. In the medium term, greater inflow volumes should increase nutrient input, which may favour cyanobacteria growth [9,65]. Greater inflow volumes should also increase turbidity in reservoirs through suspended solids input and resuspension from bottom layers, which may reduce cyanobacteria biomass due to reduced light availability, or favour buoyant cyanobacteria.

### 4.3. Mitigation Strategies

Nowadays, the technology for the treatment of point pollution from wastewater is well established. The barriers to treatment implementation mostly come from lack of funding and political commitment [19]. In turn, non-point pollution is an important source of nutrients, suspended solids [51] and other pollutants [78] to lakes and reservoirs, which is more difficult to reduce than

point sources [71]. According to Hamilton et al. [19], remediating measures for controlling N and P from non-point source pollution should include source control and the enhancement of natural processes occurring along hydrological pathways. The increase of impervious areas can be mitigated by introducing the techniques of water sensitive urban design (WSUD) in the catchment. WSUD encompasses all aspects of integrated urban water cycle management and its aim, among others, is to minimize the hydrological impacts of urban sprawl on the surrounding environment, including enhancing and maintaining water quality [79]. For this purpose, infiltration and retention techniques, such as green roofs, infiltration wells/trenches and rain gardens, are used to retain water onsite as much as possible and to promote water infiltration using natural site characteristics. Such techniques have potential to reduce suspended solids and related pollutants, total nitrogen and total phosphorus concentration in runoff through filtration, chemical sorption, biodegradation, phytoremediation, among others [80,81].

WSUD techniques may integrate nature-based solutions (NBS) such as an interconnected network of natural and artificial green spaces and water-bodies, within and between urban areas, given origin to the so called blue green infrastructures. A coupled modelling approach such as the one described in this paper would be very useful for assessing the impact of these NBS on the reservoir water quality, optimizing their location in the watershed and discussing the more appropriate socio-economic and environmental alternatives with stakeholders. Urban water bodies, green areas and urban drainage infrastructures have great potential to provide valuable ecosystem services not only related to water quality but also to air quality, increasing biodiversity, mitigating heat island development, creating opportunities for leisure, sports, social connections and diversifying income generation opportunities [82,83].

## 5. Conclusions

An integrated modelling approach coupling a catchment model to a lake ecological model was conducted in a tropical urban lake and its catchment to assess stormwater runoff impacts on cyanobacteria dynamics. Scenarios of increased imperviousness of the catchment and of a reduction in the load of nutrients and suspended solids in dry weather inflow were simulated. Reducing the load of phosphorus, the main limiting nutrient of cyanobacteria growth, in the inflow, led to a decrease in its concentration in Lake Pampulha during the study period. Among other scenarios, it had the greatest impact on cyanobacteria biomass, which was lower and whose peaks were smaller and delayed during the whole simulation period. However, even decreasing the phosphorus load into Lake Pampulha by half would not be sufficient to achieve the water quality objective of a maximum chlorophyll-a concentration of 60 µg L$^{-1}$.

Increased imperviousness in the catchment led to higher runoff volumes and loads of TP, NO$_3^-$ and TSS, while NH$_4^+$ seemed to be mainly diluted by stormwater. According to the simulated scenarios, the larger and more polluted inflow into Lake Pampulha will raise cyanobacteria biomass peaks mainly in the beginning of the wet season when additional nutrient input from catchment runoff will increase nutrient concentration in the reservoir surface layers. Model results also suggest that higher TSS input from a more impervious catchment will have temporary negative effects on cyanobacteria growth due to low light availability.

From our simulated scenarios, recovering Lake Pampulha water quality will require efforts to (1) reduce pollutant load in dry weather flow through improvement in the sewer system and; (2) to mitigate the impacts of stormwater runoff. The latter could be achieved using more sustainable and nature-based urban drainage solutions based on pollutant source control through the retention and infiltration of stormwater using natural site characteristics.

The challenges involved in runoff water quality modelling in urban catchments and in the modelling of cyanobacteria biomass dynamics in lakes and reservoirs are already widely reported in literature. Coupling these models in an integrated approach brings uncertainties which must be acknowledged for and carefully treated in future studies in order to make results from scenario

simulations more reliable. Modelling approaches which link the catchment, where runoff and pollutant loads are influenced by human activities, to lentic environments, which provide a wide range of ecosystem services, are helpful tools to cope with this uncertain future of global warming and increasing urbanisation.

**Author Contributions:** T.F.G.S., B.V.L., B.J.L. and N.O.N. elaborated the conceptual integrated modelling approach. N.O.N. obtained funding. G.P. elaborated and designed the calibration procedure for the hydrological model. A.G. elaborated the methodology for nutrients and chlorophyll-a analysis and supervised its application. C.C.F. elaborated the methodology for phytoplankton counting and identification and supervised its application. A.G. and C.C.F. gave support regarding the theoretical background on phytoplankton dynamics. G.P. and N.O.N. gave support regarding the theoretical background on hydrological modelling. B.V.L. and B.J.L. gave support regarding the theoretical background on lake modelling. T.F.G.S. conducted the field work and data analysis. T.F.G.S. wrote the manuscript. All authors contributed to the discussion of the results and the proofreading of the manuscript.

**Funding:** This research was conducted within the Stormwater Management Project (MAPLU 2 project) funded by the Brazilian agencies FINEP (Financiadora de Inovação e Pesquisa) and CNPq. A. Giani and N. Nascimento are CNPq fellows (Brazilian researcher fellowship program). CAPES and Ecole des Ponts Paris Tech funded Talita Silva's scholarship. UFMG funded publishing costs.

**Acknowledgments:** The authors are very grateful to the Belo Horizonte Municipality and to INMET for providing data. Thanks to the CAPES-COFECUB (Coordenação de Aperfeiçoamento de Pessoal de Nível Superior—Comité Français d'Évaluation de la Coopération Universitaire et Scientifique avec le Brésil) project, the mobility of researchers was made possible. We are also grateful to Viet Tran-Khac, Philippe Dubois, Mohamed Saad, Martin Seidl, Daniel Zuim, Priscila Siqueira and Elenice Silva for their support in different moments of this research.

**Conflicts of Interest:** The authors declare no conflict of interest. The funders had no role in the design of the study; in the collection, analyses, or interpretation of data; in the writing of the manuscript, or in the decision to publish the results".

## Appendix A  Hydrological Model Parameters

**Table A1.** Range of parameter values used in the calibration runoff water quality and parameter calibrated values.

| Parameter | Range | Unity | Calibrated Parameter |
|:---:|:---:|:---:|:---:|
| $F_{build-up}$ TSS | 10–200 | kg ha$^{-1}$ | 196 |
| *dec* TSS | 0–0.8 | day$^{-1}$ | 0.77 |
| $F_{build-up}$ NH$_4^+$ | 0–2 | kg ha$^{-1}$ | 0 |
| *dec* NH$_4^+$ | 0–0.01 | day$^{-1}$ | 0 |
| $F_{build-up}$ NO$_3^-$ | 0–6 | kg ha$^{-1}$ | 5.95 |
| *dec* NO$_3^-$ | 0–0.2 | day$^{-1}$ | 0.19 |
| $F_{build-up}$ TP | 0–10 | kg ha$^{-1}$ | 0.78 |
| *dec* TP | 0–0.2 | day$^{-1}$ | 0.19 |
| *w* TSS | 0–0.5 | mm$^{-1}$ | 0.02 |
| *wpo* TSS (mean) | 0–2.5 | - | 1.38 |
| *w* NH$_4^+$ | 0–0.01 | mm$^{-1}$ | 0 |
| *wpo* NH$_4^+$ | 0–2 | - | 0 |
| *w* NO$_3^-$ | 0–0.3 | mm$^{-1}$ | 0.005 |
| *wpo* NO$_3^-$ (mean) | 0–2 | - | 0.93 |
| *w* TP | 0–0.1 | mm$^{-1}$ | 0.02 |
| *wpo* TP (mean) | 0–2 | - | 1.08 |

## Appendix B  Phytoplankton Model Equations

**Table A2.** Generic parameterizations of processes modelled in CAEDYM. Equations for light and water temperature dependency. a is related to phytoplankton group.

| Equation |
|---|
| (a)  Light |
| $$I(z) = K_{par} \times I_0 e^{-k_D z} \tag{A1}$$ |
| $$k_D = k_w + K_{eDOC} DOC + K_{ePOC} POC + \\ + K_{eSS} SS \sum_{a=1}^{N} K_e^{A_a}(A_a) \tag{A2}$$ |
| (b)  Temperature dependency (c, d and e are internally calculated) |
| $$f(T)^1 = \vartheta^{T-20} - \vartheta^{c(T-d)} + e$$ |
| $$T = T_{std}: \quad f(T) = 1$$ |
| $$T = T_{opt}: \quad \frac{\partial f(T)}{\partial t} = 0 \tag{A3}$$ |
| $$T = T_{max}: \quad f(T) = 0$$ |
| $$T < T_{std}: \quad f(T)^2 = \vartheta^{T-20} \tag{A4}$$ |

**Table A3.** Equations of phytoplankton modelling in CAEDYM. a is related to phytoplankton group.

| Equation |
|---|
| (a)  Phytoplankton growth ($\mu_a$) and loss due to mortality and respiration ($L_a$) and settling ($M_{Va}$) |
| $$\mu_a = \mu_{maxa} \times min[f(I),\ f(N),\ f(P)] \times f(T) \tag{A5}$$ |
| $$L_a = k_{ra} \times f_a^{T2}(T) + k_{rpa} \times \mu_a \tag{A6}$$ |
| $$M_{V_a} = \frac{V_{s_a}}{\Delta z} \tag{A7}$$ |
| (b)  Carbon uptake through photosynthesis ($U$) and loss through respiration (R) and through mortality and excretion ($E$) |
| $$U_{CO_2}(A_a) = \mu_a \times \left(1 - k_{ptf}\right) \times Y_{c:chla} \tag{A8}$$ |
| $$R_{DIC}(A_a) = \left[L_{g_a} \times f_{RES_a} + k_{rpa} \times \mu_a\right] \times Y_{c:chla} \tag{A9}$$ |
| $$E_{DOC}(A_a) = \left\{\left[\left(1 - f_{RES_a}\right) \times L_a\right] \times Y_{c:chla}\right\} \times f_{DOM_a} \tag{A10}$$ |
| $$E_{POC}(A_a) = \{[(1 - f_{RESa}) \times L_a] \times Y_{c:chla}\} \times \left(1 - f_{DOM_a}\right) \tag{A11}$$ |
| (c)  Light limitation |
| $$f(I) = 1 - exp\left(\frac{I}{I_K}\right) \tag{A12}$$ |
| Nitrogen limitation $f(N)$, uptake $U$ and loss $E$ |
| $$f(N) = \frac{AIN_{maxa}}{AIN_{maxa} - AIN_{mina}} \times \left[1 - \frac{AIN_{mina}}{AIN_a}\right] \tag{A13}$$ |
| $$U_{NH_4}(A_a) = UN_{max_a} \times P_{N_a} \times \left[f_{A_a}^{T1}(T) \times \frac{AIN_{maxa} - AIN_a}{AIN_{maxa} - AIN_{mina}} \times \frac{NO_3 + NH_4}{NO_3 + NH_4 + K_{N_a}}\right] \tag{A14}$$ |
| $$U_{NO_3}(A_a) = \\ UN_{max_a} \times (1 - P_{N_a} \times \left[f_{A_a}^{T1}(T) \times \frac{AIN_{maxa} - AIN_a}{AIN_{maxa} - AIN_{mina}} \times \frac{NO_3 + NH_4}{NO_3 + NH_4 + k_{N_a}}\right] \tag{A15}$$ |
| $$P_N = \frac{NH_4 \times NO_3}{(NH_4 + k_{Na})} + \frac{NH_4 \times k_N}{(NH_4 + NO_3)(NO_3 + k_{Na})} \tag{A16}$$ |
| $$E_{DON}(A_a) = \left[\frac{AIN_a}{A_a} \times L_a\right] \times f_{DOM_a} \tag{A17}$$ |
| $$E_{PON}(A_a) = \left[\frac{AIP_a}{A_a} \times L_a\right] \times \left(1 - f_{DOM_a}\right) \tag{A18}$$ |
| Phosphorus limitation $f(P)$, uptake $U$ and loss $E$ |
| $$f(P) = \frac{AIP_{maxa}}{AIP_{maxa} - AIP_{mina}} \times \left[1 - \frac{AIP_{mina}}{AIP_a}\right] \tag{A19}$$ |
| $$U_{FRP}(A_a) = UP_{max_a}\left[f_{A_a}^{T1}(T) \times \frac{AIP_{maxa} - AIP_a}{AIP_{maxa} - AIP_{mina}} \times \frac{FRP}{FRP + k_{P_a}}\right] \tag{A20}$$ |
| $$E_{DOP}(A_a) = \left[\frac{AIP_a}{A_a} \times L_a\right] \times f_{DOM_a} \tag{A21}$$ |
| $$E_{POP}(A_a) = \left[\frac{AIP_a}{A_a} \times L_a\right] \times \left(1 - f_{DOM_a}\right) \tag{A22}$$ |

**Table A4.** Description of CAEDYM variables for phytoplankton modelling. a is related to phytoplankton group.

| Description | Symbol |
|---|---|
| Ammonium | $NH_4^+$ |
| Computational time step | $\Delta t$ |
| Depth | z |
| Detrital particulate organic carbon concentration | POC |
| Detrital particulate organic nitrogen concentration | PON |
| Detrital particulate organic phosphorus concentration | POP |
| Dissolved inorganic carbon | DIC |
| Dissolved organic carbon concentration | DOC |
| Dissolved organic matter | DOM |
| Dissolved organic nitrogen concentration | DON |
| Dissolved organic phosphorus concentration | DOP |
| Filterable reactive phosphorus | FRP |
| Light extinction coefficient | $k_D$ |
| Light intensity (photosynthetically active radiation - PAR) | I |
| Loss rate | $L_a$ |
| Filterable reactive phosphorus | FRP |
| Time index | i |
| Incident shortwave intensity at water surface | $I_0$ |
| Internal nitrogen concentration | $AIN_a$ |
| Internal phosphorus concentration | $AIP_a$ |
| Nitrate | $NO_3^-$ |
| Nitrogen | N |
| pH | pH |
| Phosphorus | P |
| Phytoplankton biomass | A |
| Phytoplankton group index | a |
| Phytoplankton loss due to settling | $M_{Va}$ |
| Suspended solids | SS |
| Vertical thickness of computational cell | $\Delta z$ |
| Water temperature | T |

**Table A5.** Description of CAEDYM parameters for phytoplankton modelling. A is related to phytoplankton group.

| Description | Symbol |
|---|---|
| (a) Light | |
| Fraction of incoming solar radiation which is photosynthetically active | $K_{par}$ |
| Specific light attenuation coefficient due to the action of pure water | $K_W$ |
| Specific light attenuation coefficient rate due to the action of DOC | $K_{eDOC}$ |
| Specific light attenuation coefficient rate due to the action of POC | $K_{ePOC}$ |
| Specific light attenuation coefficient rate due to the action of SS | $K_{eSS}$ |
| Specific light attenuation coefficient rate due to Phytoplankton | $K_e^A$ |
| (b) Temperature dependency | |
| Arrhenius constant | $\vartheta$ |
| Standard temperature | $T_{std}$ |
| Optimum temperature | $T_{opt}$ |
| Maximum temperature | $T_{max}$ |
| (c) Phytoplankton growth | |
| Maximum potential growth rate | $\mu_{maxa}$ |
| Phytoplankton growth rate | $\mu_a$ |
| Metabolic loss rate coefficient | $k_{ra}$ |
| Fraction of phytoplankton production lost due to photorespiration | $k_{rpa}$ |
| Settling velocity | $V_{sa}$ |

**Table A5.** *Cont.*

| Description | Symbol |
| --- | --- |
| (d) Carbon uptake and loss | |
| Stoichiometric ratio of C to *chla* | $Y_{C:chla}$ |
| Fraction of respiration relative to total metabolic loss | $f_{RES}$ |
| Fraction of metabolic loss rate that goes to DOM | $f_{DOMa}$ |
| Fraction of photorespiration | $k_{ptr}$ |
| (e) Light limitation | |
| Light intensity for maximum phytoplankton production | $I_k$ |
| (f) Nitrogen uptake and loss | |
| Half saturation constant for nitrogen uptake | $K_{Na}$ |
| Maximum internal nitrogen concentration | $AIN_{maxa}$ |
| Maximum rate of nitrogen uptake | $UN_{max}$ |
| Minimum internal nitrogen concentration | $AIN_{mina}$ |
| Phytoplankton group preference for $NH_4^+$ | $P_{Na}$ |
| (g) Phosphorus uptake and loss | |
| Half saturation constant for phosphorus uptake | $K_{Pa}$ |
| Maximum internal phosphorus concentration | $AIP_{maxa}$ |
| Maximum rate of phosphorus uptake | $UP_{maxa}$ |
| Minimum internal phosphorus concentration | $AIP_{mina}$ |

## Appendix C Phytoplankton Model Calibrated Parameters

**Table A6.** Range of parameter values used in the calibration of phytoplankton modelling parameter calibrated values.

| Parameter | Symbol | Unity | Range | Calibrated Value |
| --- | --- | --- | --- | --- |
| (a) Cyanobacteria (index C) | | | | |
| Maximum potential growth rate | $\mu_{maxC}$ | day$^{-1}$ | 0.25–1.60 | 1.60 |
| Metabolic loss rate coefficient | $k_{RC}$ | day$^{-1}$ | 0.05–0.15 | 0.05 |
| Respiration temperature dependency | $\vartheta_{RC}$ | - | 1.04–1.10 | 1.05 |
| Maximum internal P concentration | $AIP_{maxC}$ | mg P (mg chl-a)$^{-1}$ | 0.92–3.80 | 0.92 |
| Maximum rate of P uptake | $UP_{maxC}$ | mg P (mg chl-a)$^{-1}$ day$^{-1}$ | 0.4–54.4 | 24.4 |
| Maximum rate of N uptake | $UN_{maxC}$ | mg N (mg chl-a)$^{-1}$ day$^{-1}$ | 0.2–4.8 | 0.71 |
| Optimum temperature | $T_{optC}$ | °C | 25–35 | 29 |
| Specific attenuation coefficient | $k_e{}^C$ | (mg chl-a L$^{-1}$) m$^{-1}$ | 0.01–0.02 | 0.02 |
| Light intensity for maximum phytoplankton production | $I_{kC}$ | μEm$^{-2}$ s$^{-1}$ | 15–180 | 33 |
| (b) Other phytoplankton (index O) | | | | |
| Maximum potential growth rate | $\mu_{maxO}$ | day$^{-1}$ | 0.50–1.84 | 0.50 |
| Metabolic loss rate coefficient | $k_{RO}$ | day$^{-1}$ | 0.02–0.12 | 0.03 |
| Respiration temperature dependency | $\vartheta_{RO}$ | - | 1.04–1.12 | 1.10 |
| Maximum rate of P uptake | $UP_{maxO}$ | mg P (mg chl-a)$^{-1}$ day$^{-1}$ | 1.0–4.5 | 3.3 |
| Optimum temperature | $T_{optO}$ | °C | 21–29 | 26 |
| Maximum temperature | $T_{maxO}$ | °C | 30–35 | 33 |
| Specific attenuation coefficient | $k_e{}^O$ | (mg chl-a L$^{-1}$) $^{-1}$ m$^{-1}$ | 0.01–0.02 | 0.012 |
| Light intensity for maximum phytoplankton production | $I_{kO}$ | μEm$^{-2}$ s$^{-1}$ | 20–250 | 173 |
| (c) Sediment parameter | | | | |
| Static sediment exchange rate | $r_{SOS}$ | g m$^{-2}$ day$^{-1}$ | 0.92–7.97 | 1.7 |

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
