# Peer review of "Impact of Urban Stormwater Runoff on Cyanobacteria Dynamics in A Tropical Urban Lake"

_water, doi:10.3390/w11050946_

Round 1
Reviewer 1 Report
The article "Urban stormwater runoff impacts on cyanobacteria
dynamics in a tropical urban lake" by Talita Silva et al., is very well prepared. The applied mathematical models, their setup, calibration and validation do not raise any objections, but in my opinion, when discussing the results, little attention was paid to the problem of internal supply in nutrients (especially phosphorus) from bottom sediments, while it is probably the main source of phosphorus for phytoplankton primary production in the dry season of the year.The proposed reduction in external loading is of course very important, but without limiting the internal source of phosphorus it will not be possible to achieve the assumed effects of low chlorophyll-a and elimination of cyanobacterial water bloom. It is necessary to plan the restoration of the reservoir, preferably using sustainable restoration treatments simultaneously applying few nature-based methods.Discussing this problem in the text will require adding a paragraph in the discussion section and one sentence in the conclusions.
Author Response
Point 1. The article "Urban stormwater runoff impacts on cyanobacteria dynamics in a tropical urban lake" by Talita Silva et al., is very well prepared. The applied mathematical models, their setup, calibration and validation do not raise any objections, but in my opinion, when discussing the results, little attention was paid to the problem of internal supply in nutrients (especially phosphorus) from bottom sediments, while it is probably the main source of phosphorus for phytoplankton primary production in the dry season of the year. The proposed reduction in external loading is of course very important, but without limiting the internal source of phosphorus it will not be possible to achieve the assumed effects of low chlorophyll-a and elimination of cyanobacterial water bloom. It is necessary to plan the restoration of the reservoir, preferably using sustainable restoration treatments simultaneously applying few nature-based methods. Discussing this problem in the text will require adding a paragraph in the discussion section and one sentence in the conclusions.
Response 1: Nowadays, the main source of phosphorus into Lake Pampulha during dry weather is domestic wastewater discharges that come into the reservoir through the tributaries. Figure 10.b that we added in this revised version, shows that even reducing the total phosphorus input into the lake by half, PO43- concentration in the surface layer remains higher than 0.02 mgP.L-1 most of the time. Figure 8 also shows that the phosphorus input from the catchment is large even during dry weather. However, we agree that when point and non-point pollutant loading in the catchment will be reduced, the internal load may become a major concern and impair the efforts to reduce point and non-point pollution. In particular, we think that water resource managers should be aware of this possibility. We added information about it in the Discussion section (lines 612-616).
Reviewer 2 Report
Manuscript number: water-475342
Title: Urban stormwater runoff impacts on cyanobacteria 2 dynamics in a tropical urban lake
The manuscript presents results of the use of lake ecological modelling to predict the impact of storm water on water quality in lakes. The Authors proposed integrated modelling approach coupling hydrological model (SWMM) and lake ecological model (DYRESM-CAEDYM) to assess how changes in the catchment can impact algal biomass in lake ecosystem. The manuscript fits with the journal aims. However, some important questions and doubts should be clarified before acceptance of the manuscript.
The objective of this paper “…present a modelling tool for assessing the link between cyanobacteria dynamics in an urban lake and changes in the catchment” can`t be realized because of the lack of row cyanobacteria data for the presented analysis
In Materials and methods there is only a statement that “…DYCD model was set up to explicitly simulate cyanobacteria, in view of their dominance during most of the monitoring period in Lake Pampulha…”
How the biomass of cyanobacteria was determined? Concentration of chlorophyll-a is an indicator of total phytoplankton biomass, not a single taxonomic group.
Also I recommend English editing of the manuscript by native speaker
Author Response
Point 1. The manuscript presents results of the use of lake ecological modelling to predict the impact of storm water on water quality in lakes. The Authors proposed integrated modelling approach coupling hydrological model (SWMM) and lake ecological model (DYRESM-CAEDYM) to assess how changes in the catchment can impact algal biomass in lake ecosystem. The manuscript fits with the journal aims. However, some important questions and doubts should be clarified before acceptance of the manuscript.
The objective of this paper “…present a modelling tool for assessing the link between cyanobacteria dynamics in an urban lake and changes in the catchment” can`t be realized because of the lack of row cyanobacteria data for the presented analysis. In Materials and methods there is only a statement that “…DYCD model was set up to explicitly simulate cyanobacteria, in view of their dominance during most of the monitoring period in Lake Pampulha…”
How the biomass of cyanobacteria was determined? Concentration of chlorophyll-a is an indicator of total phytoplankton biomass, not a single taxonomic group.
Response 1: Actually, in the “Materials and methods” section, the description of the method for determining the phytoplankton biomass, especially the cyanobacteria biomass, was missing. During the study period, phytoplankton counting and biovolume estimation were performed under the microscope. This was added in the revised version of the paper (lines 165-167). We explained better our calibration (lines 255-258).
We included the results of phytoplankton composition in the Results (new Figure 4 and lines 352-355, 363-365, 366-368).
Most of the time, the phytoplankton biomass is dominated by cyanobacteria which represent nearly 90% of the total biomass. Therefore, the biological model was calibrated using total phytoplankton data. We then verified that cyanobacteria biomass was correctly simulated, though other phytoplankton groups were not. In the scenario simulations, the simulated cyanobacteria biomass only was considered, not the simulated total phytoplankton. We made this clearer in the revised version (lines 425-427).
Point 2. Also I recommend English editing of the manuscript by native speaker
Response 2: We asked our colleagues who have more experience in English writing, though they are not native speakers, to review our manuscript.
Reviewer 3 Report
This manuscript by author provides an interesting subject about urban stormwater runoff impacts on cyanobacteria dynamics in a tropical urban lake General Comments and I can say the same things about the manuscript. Detailed comments: 1. What is innovation of this M.S., author should give the more innovation clearly in introduction part? 2. Scenario simulation suggests that TP reduction will lead to cyanobacteria biomass decrease and to the delay of its peaks, why? authors should give more other information? 3. Increased imperviousness in the catchment will raise runoff volume, TSS, TP and NO3- load into Lake Pampulha promoting greater, cyanobacteria biomass, mainly in the beginning of the wet season why? 4. Recovering Lake Pampulha water quality will need strategies focusing on point pollution and more sustainable and nature-based solutions for urban drainage., which is important, if can give more specific information.
Author Response
This manuscript by author provides an interesting subject about urban stormwater runoff impacts on cyanobacteria dynamics in a tropical urban lake. General Comments and I can say the same things about the manuscript. Detailed comments:
Point 1. What is innovation of this M.S., author should give the more innovation clearly in introduction part?
Response 1: In this paper, we investigated how catchment urbanization impacts cyanobacteria dynamics in urban reservoirs. Research has mainly focused on large lakes, while urban reservoirs and their catchments, especially in tropical regions, are still poorly studied. Moreover, as far as we are aware, this is the first time that the impact of increased catchment imperviousness on cyanobacteria dynamics is assessed using a coupled modelling approach. We modified the abstract and the introduction in order to highlight the innovative character of this work by adding lines 17 to 22 in the abstract, and lines 47-50; 56-57; 60-62; 68-70, 79-81, 92-95 and 98-100 in the introduction.
Point 2. Scenario simulation suggests that TP reduction will lead to cyanobacteria biomass decrease and to the delay of its peaks, why? Authors should give more other information?
Response 2: In scenarios with phosphorus reduction, simulated cyanobacteria biomass was smaller and its peaks seemed to occur later, due to the smaller phosphorus (particularly PO43-) concentration in the lake surface layer. We added Figure 10.b to show the simulated PO43- concentration in the surface layer in different scenarios. In scenarios with phosphorus reduction, mean PO43- and TP concentrations in the surface layer (from 0 to 2.5 m depth) dropped by half and were significantly smaller than in other scenarios (p-value < 0.00001). We added this information in the Results section: lines 469-477, in the Discussion section (lines 598-600) and added a sentence in the abstract (line 33).
In the “Materials and methods” section, we also explained better the design of our scenarios, including phosphorus reduction scenario (line 297-303).
Point 3. Increased imperviousness in the catchment will raise runoff volume, TSS, TP and NO3- load into Lake Pampulha promoting greater, cyanobacteria biomass, mainly in the beginning of the wet season why?
Response 3: Greater stormwater volume from a more impervious catchment seems to promote cyanobacteria growth in Lake Pampulha, especially in the beginning of the wet season when PO43- and NO3- concentrations in the lake surface layer are low. During dry weather, PO43- and NO3- concentrations decrease in the surface layer due to cyanobacteria uptake. With the beginning of the rain period, additional nutrient input from catchment runoff replenishes the lake surface layer. This may provide favourable conditions for cyanobacteria blooms if the TSS input remains low. Indeed, in Lake Pampulha, a high TSS concentration in the surface layer prevents light penetration and may delay cyanobacteria growth or even contribute their decay.
To illustrate this effect, we added a new figure (Figure 11) in the Results section. Figure 11 presents daily rainfall, flow, PO43- and TSS loads into Lake Pampulha from October 25, 2011 to November 29, 2011 (beginning of the wet season). The in-lake variables which are significantly and strongly correlated to the mean cyanobacteria biomass during this period, are also presented, averaged from 0 to 2.5 m depth. We explained Figure 11 in lines 442-468. We also supported our explanation by presenting more literature results in the Discussion section (lines 623-628 and 632-642).
We explained this research outcome in the Conclusion (lines 695-699) and in the Abstract (line 34-35).
Point 4. Recovering Lake Pampulha water quality will need strategies focusing on point pollution and more sustainable and nature-based solutions for urban drainage, which is important, if can give more specific information.
Response 4: Sustainable and nature-based solutions must be promoted as remediating measures for controlling N and P from non-point source pollution. They should include source control and the enhancement of natural processes occurring along hydrological pathways. Examples are green roofs, infiltration wells/trenches or rain gardens. These techniques may improve stormwater quality through the filtration, retention and sedimentation of particulate matter, chemical sorption, biodegradation, phytoremediation ... Integrated in an interconnected network of natural and artificial green areas and water bodies, they will improve stormwater quality and provide varied ecosystem services. We tried to highlight this point by adding some sentences and references in the Discussion section (lines: 655-656, 659-6561, 669-671) and a sentence at the abstract end (lines 40-42).
Round 2
Reviewer 2 Report
Manuscript number: water-475342
Title: Urban stormwater runoff impacts on cyanobacteria dynamics in a tropical urban lake
Responses to revised version
The Authors responded clearly to the comments included in revision. The manuscript was significantly improved. I recommend its publication in Water.
Reviewer 3 Report
afte revion this M.S can be publicaiotn.